# Mechanism of self/nonself-discrimination in *Brassica* self-incompatibility

Kohji Murase[1,10], Yoshitaka Moriwaki [2,3,10 ✉], Tomoyuki Mori [4,10], Xiao Liu[4], Chiho Masaka[4], Yoshinobu Takada[5], Ryoko Maesaki[6], Masaki Mishima[6], Sota Fujii [1], Yoshinori Hirano [4,9], Zen Kawabe[1], Koji Nagata [1], Tohru Terada [2,3,7], Go Suzuki [8], Masao Watanabe [5], Kentaro Shimizu [2,3], Toshio Hakoshima [4 ✉] & Seiji Takayama [1 ✉]

Self-incompatibility (SI) is a breeding system that promotes cross-fertilization. In *Brassica*, pollen rejection is induced by a haplotype-specific interaction between pistil determinant SRK (*S* receptor kinase) and pollen determinant SP11 (*S*-locus Protein 11, also named SCR) from the *S*-locus. Although the structure of the *B. rapa* $S_9$-SRK ectodomain (eSRK) and $S_9$-SP11 complex has been determined, it remains unclear how SRK discriminates self- and nonself-SP11. Here, we uncover the detailed mechanism of self/nonself-discrimination in *Brassica* SI by determining the $S_8$-eSRK–$S_8$-SP11 crystal structure and performing molecular dynamics (MD) simulations. Comprehensive binding analysis of eSRK and SP11 structures reveals that the binding free energies are most stable for cognate eSRK–SP11 combinations. Residue-based contribution analysis suggests that the modes of eSRK–SP11 interactions differ between intra- and inter-subgroup (a group of phylogenetically neighboring haplotypes) combinations. Our data establish a model of self/nonself-discrimination in *Brassica* SI.

[1] Department of Applied Biological Chemistry, Graduate School of Agricultural and Life Sciences, The University of Tokyo, Tokyo 113-8657, Japan. [2] Department of Biotechnology, Graduate School of Agricultural and Life Sciences, The University of Tokyo, Tokyo 113-8657, Japan. [3] Collaborative Research Institute for Innovative Microbiology, The University of Tokyo, Tokyo 113-8657, Japan. [4] Graduate School of Biological Sciences, Nara Institute of Science and Technology, Nara 630-0192, Japan. [5] Graduate School of Life Sciences, Tohoku University, Sendai 980-8577, Japan. [6] Graduate School of Science, Tokyo Metropolitan University, Tokyo 192-0397, Japan. [7] Agricultural Bioinformatics Research Unit, Graduate School of Agricultural and Life Sciences, The University of Tokyo, Tokyo 113-8657, Japan. [8] Division of Natural Science, Osaka Kyoiku University, Kashiwara 582-8582, Japan. [9] Present address: Graduate School of Pharmaceutical Sciences, The University of Tokyo, Tokyo 113-0033, Japan. [10] These authors contributed equally: Kohji Murase, Yoshitaka Moriwaki, Tomoyuki Mori. ✉email: moriwaki@bi.a.u-tokyo.ac.jp; hakosima@bs.naist.jp; a-taka@g.ecc.u-tokyo.ac.jp

Many flowering plants promote outbreeding by self-incompatibility (SI) to maintain their genetic diversity[1]. In *Brassica*, self/nonself-discrimination in the SI reaction is sporophytically controlled by a single locus, called *S*, with over 100 haplotypes ($S_1$, $S_2$, …, $S_n$)[2–4]. Once the *S*-haplotypes in pollen and pistil are matched, the pollen is rejected on the papilla cell surface of pistil. To date, five genes involved in SI have been found in the *S*-locus. The first *S*-locus product to be identified is a secreted glycoprotein, *S*-locus glycoprotein (SLG). SLG consists of two lectin domains, an EGF-like domain, and a PAN domain, and enhances the pistil-side SI reaction by unknown mechanism[5–7]. *S* receptor kinase (SRK), a receptor kinase containing an SLG-like ectodomain, transmembrane region, and intracellular kinase domain, acts as the pistil determinant[7,8]. *S*-locus Protein 11 (SP11; also called SCR), a secreted small basic protein with four disulfide bonds forming a defensin-like structure, functions as the pollen determinant[9–13]. Two small RNAs, *SP11 methylation inducer* (*SMI*) and *SMI2*, regulate the dominant–recessive hierarchy on the pollen side by suppressing the *SP11* expression[14,15].

Pistil factor SRK, localized on the plasma membrane of the papilla cell, specifically recognizes its cognate pollen factor SP11 released from the pollen surface, triggering self-phosphorylation of the kinase domain[16–19], and transduces the SI signal to intracellular downstream effectors, resulting in pollen rejection[20–25]. Both SRK and SP11 are highly polymorphic proteins among the haplotypes, with three hypervariable (HV I–III) regions in SRK; SP11 proteins have few similarities except for the signal sequences and cysteines forming the disulfide bonds[26,27]. These sequence variations are thought to be important for self/nonself-discrimination in *Brassica* SI; however, the discrimination mechanism is still largely unclear.

The recent determination of the complex structure of the $S_9$-SRK ectodomain (eSRK) and $S_9$-SP11[9,28] (called eSRK9–SCR9 in Ma et al.[29]) revealed the mechanism of $S_9$-SP11 recognition by $S_9$-SRK in *B. rapa* SI. However, the mechanisms of ligand recognition in other haplotypes and self/nonself-discrimination remained unknown. Here, we report the crystal structure of engineered $S_8$-eSRK and $S_8$-SP11 complex derived from $S_8$-haplotype in *B. rapa*, which has a common overall structure with the $S_9$-eSRK–$S_9$-SP11 complex but an entirely different mode of the ligand recognition. Our comprehensive interaction analysis of modeled and known eSRK and SP11 structures by molecular dynamics (MD) simulations reveals the important features for understanding the mechanism of self/nonself-discrimination in *Brassica* SI.

## Results

**Structure of the $S_8$-eSRK–$S_8$-SP11 complex**. To further understand the mechanism by which SRK discriminates self- or nonself-SP11 in *Brassica* SI, we tried to determine the structure of the *B. rapa* $S_8$-eSRK–$S_8$-SP11[10,30] complex. $S_8$-eSRK (residues 32–433) expressed in insect cells by a baculovirus system was highly aggregated (Fig. 1a). Comprehensive protein engineering experiments yielded $S_8$-eSRK containing 11 mutations ($S_8$-meSRK) in two lectin domains that improved the expression of recombinant protein in the insect cell system (Fig. 1a; Supplementary Fig. 1a, b). Pull-down, isothermal calorimetry (ITC), gel-filtration, and chemical shift perturbation (CSP) analysis monitored by $^1$H-$^{15}$N HSQC spectra revealed that $S_8$-meSRK still strongly bound $S_8$-SP11 (Fig. 1b–d; Supplementary Fig. 1c–e). In the pull-down assay, $S_8$-meSRK-HLH, but not $S_8$-meSRK, bound $S_8$-SP11, consistent with our previous experiments[18]. $S_8$-meSRK ($S_8$-eSRK) seems difficult to form the ligand-receptor complex in the environment with low concentration of $S_8$-meSRK such as the pull-down assay, in contrast to the high concentration conditions

in ITC, gel-filtration, and CSP experiments. The dimerization domain (HLH) in $S_8$-meSRK-HLH is supposed to enhance $S_8$-meSRK–$S_8$-SP11 interaction by supporting the SP11–induced SRK dimerization.

The crystal structure of the $S_8$-meSRK ectodomain and chemically synthesized $S_8$-SP11 complex was determined at 2.6 Å resolution. The complex, composed of a 2:2 $S_8$-meSRK–$S_8$-SP11 heterotetramer, forms a turned A-like structure (Fig. 1e). The structures of symmetrical molecules in a single complex are almost the same in both $S_8$-meSRK and $S_8$-SP11, with the root mean squared deviation (rmsd) values of 0.35 and 0.34 Å, respectively (Supplementary Fig. 2a, b). $S_8$-meSRK consists of two lectin domains, an EGF-like domain, and a PAN domain with six disulfide bonds and three sugar chains (Supplementary Fig. 2c). Two SP11 molecules bind to the $S_8$-meSRK surface comprising three HV regions[26] whose sequences vary significantly among *Brassica* S-haplotypes (Supplementary Fig. 2d). These features are similar to those of the $S_9$-eSRK–$S_9$-SP11 complex structure[29], assuming that the overall structures of the eSRK–SP11 complexes in other S-haplotypes are basically conserved.

Although the overall rmsd value between the $S_8$-meSRK–$S_8$-SP11 and $S_9$-eSRK–$S_9$-SP11 complexes is 2.11 Å, when single eSRK molecules of the two S-haplotypes are superimposed (rmsd = 0.88 Å), the opposite sides of the eSRK molecules exhibit six-degree torsion against the central axis of the complexes (Supplementary Fig. 3a, b). Superimposition of $S_8$-SP11 and $S_9$-SP11 (rmsd = 2.29 Å) reveals similar β-sheet structures, but different lengths and angles in the α-helix structures (Supplementary Fig. 3c). In the two SP11-binding sites of $S_8$-meSRK, site 1 is 793.0 Å$^2$ of the concaved surface area formed by all HV regions, and site 2 is a 444.1 Å$^2$ crescent moon–like surface mainly formed by the HV II region (Supplementary Fig. 3d). These contact areas are similar to $S_9$-eSRK (site 1, 782.6 Å$^2$; site 2, 429.7 Å$^2$); however, the shapes are slightly different (Supplementary Fig. 3d). These differences likely contribute to self-/nonself-discrimination ability.

**SP11 recognition in $S_8$- and $S_9$-complexes**. The β2–β3 loop of $S_8$-SP11 forms a hydrogen bond network with the SP11-binding site 1 of $S_8$-meSRK (Fig. 2a). The amino group of Lys63 in $S_8$-SP11, located on the highly acidic concave surface of $S_8$-meSRK, stabilizes the interaction. Phe69 of $S_9$-SP11, which corresponds to Lys63 of $S_8$-SP11, interacts with the hydrophobic concaved surface of $S_9$-eSRK (Fig. 2b, c; Supplementary Fig. 3e). Met64 of $S_8$-SP11 forms a contact alongside the main chain atoms between residues 296 and 297 of $S_8$-meSRK and is solely involved in a homodimeric $S_8$-SP11–$S_8$-SP11 contact with a small contact area (52.1 Å$^2$), in contrast to the absence of a homodimeric SP11–SP11 contact in the $S_9$-complex (Fig. 2a–c; Supplementary Fig. 3f). Interestingly, only the side chain of Phe290 in $S_9$-eSRK (His297 in $S_8$-meSRK) flips to the SP11 molecule and engages in the steric clash against the Met64 of $S_8$-SP11 when the eSRK molecules are superimposed (Fig. 2a–c). Met301 and Met304 of $S_8$-meSRK also stabilize the complex by hydrophobic interactions with the residues in the β2–β3 loop and β1 strand of $S_8$-SP11 (Supplementary Fig. 3f). Arg303 of $S_8$-meSRK forms three hydrogen bonds with $S_8$-SP11, whereas Met296 in the same position of $S_9$-eSRK contributes to binding through hydrophobic interactions (Fig. 2d–f). Phe34 and Lys36 in the β1 strand of $S_8$-SP11 interact with the HV III residues (Ile339 and Asn337) of $S_8$-meSRK, whereas the α1 helix of $S_9$-SP11 mainly contacts HV III in the $S_9$-complex (Fig. 2d–f).

In SP11-binding site 2, the aromatic ring of Tyr275 in $S_8$-meSRK is stacked with the plane of Asn65 side chain in $S_8$-SP11, which forms three hydrogen bonds with Asn271, Glu277, and

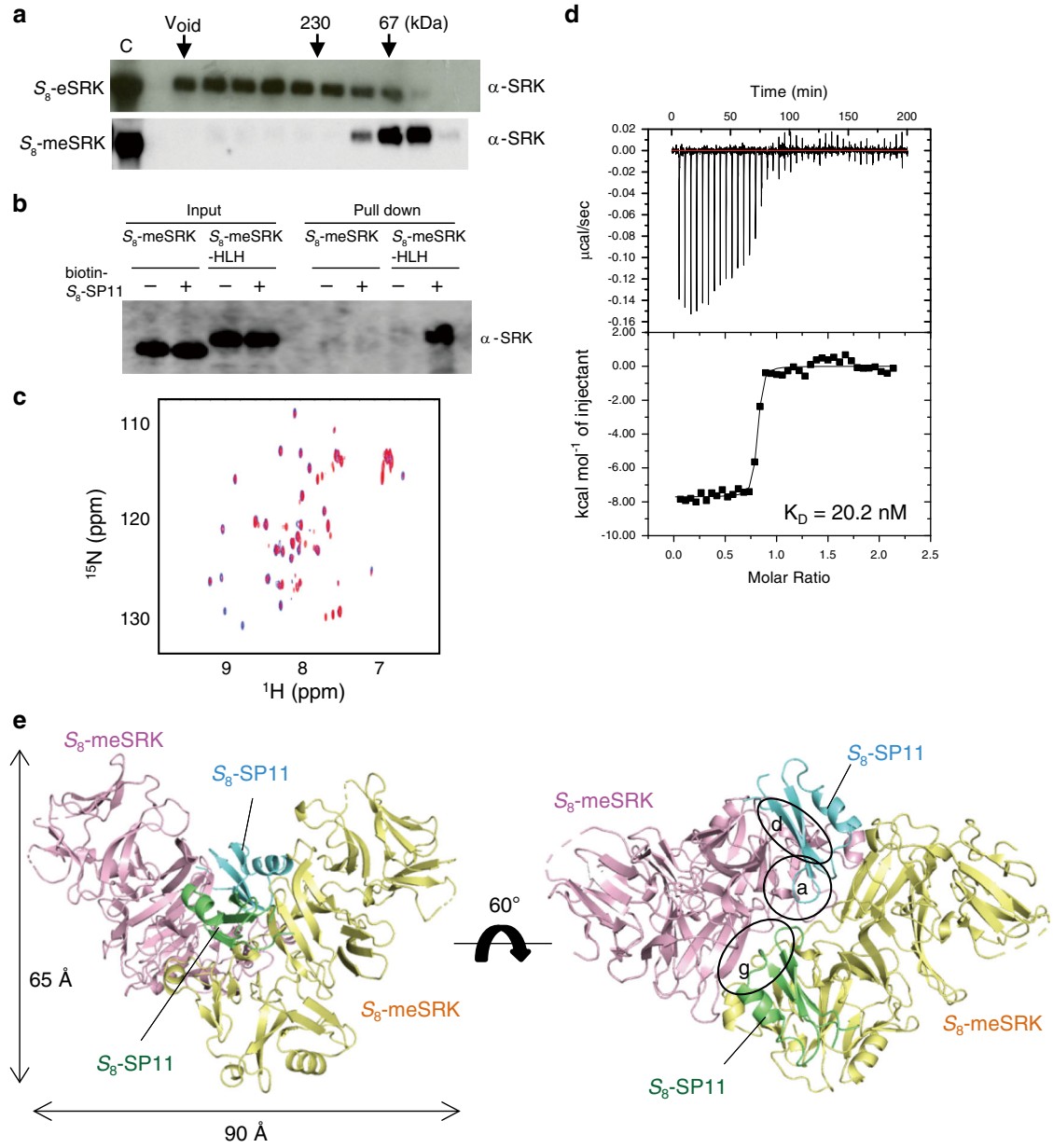

**Fig. 1 Structure determination of $S_8$-meSRK–$S_8$-SP11 complex. a** Eleven amino acid mutations of $S_8$-eSRK ($S_8$-meSRK) suppress self-aggregation. The immunoblots show the fractions from size exclusion chromatography of $S_8$-eSRK and $S_8$-meSRK ectodomains expressed in insect cells. Arrows show the elution positions of molecular mass markers. C, control supernatants of insect cell culture expressing the recombinant proteins. **b** $S_8$-meSRK possesses $S_8$-SP11 recognition activity. The panel shows immunoblot analysis of pull-down eluates using biotin-$S_8$-SP11 and insect culture media expressing $S_8$-meSRK or $S_8$-meSRK-HLH, which is an artificial fusion with the dimerization domain of a bHLH-ZIP protein[18]. **c** Chemical shift perturbation analysis of $S_8$-SP11. Overlay of the spectrum of [15]N-labeled $S_8$-SP11 (blue) with that of [15]N-labeled $S_8$-SP11 co-existing with unlabeled $S_8$-meSRK (red). **d** ITC analysis of the $S_8$-meSRK–$S_8$-SP11 interaction. Upper panel, thermogram; lower panel, integrated titration curve. **e** Overall structure of $S_8$-meSRK–$S_8$-SP11 heterotetramer. Two $S_8$-meSRK (pink and yellow) and two $S_8$-SP11 (cyan and green) molecules are shown as cartoon models. Dotted lines indicate disordered regions. Details of the labeled circles are shown in Fig. 2a, d, g.

Ser293 of $S_8$-meSRK (Fig. 2g). By contrast, no hydrogen bond and strong van der Waals interactions are found in this location on the $S_9$-complex due to a 5-Å shift of the $S_9$-SP11 β2–β3 loop from the center of the complex (Fig. 2g–i). Instead, the long α1 helix of $S_9$-SP11 forms a larger contact area than $S_8$-SP11 (Supplementary Fig. 3g). The recognition modes of symmetrical $S_8$-SP11–$S_8$-meSRK heterodimers are similar, but small differences are observed (Supplementary Fig. 3h, i). Although five of the eleven residues mutated in $S_8$-meSRK contact $S_8$-SP11, there is no negative effect on $S_8$-SP11 recognition. The importance of the $S_8$-meSRK amino acids involved in $S_8$-SP11 recognition was

confirmed by pull-down assay (Fig. 2j). The contact amino acids against $S_8$-SP11 in $S_8$-meSRK are almost in the three HV regions. Among the 31 contact residues, 21 positions are common in $S_9$-eSRK; however, only six amino acids are conserved (Supplementary Fig. 4a). The positions of the contact residues of $S_8$-SP11 against $S_8$-meSRK are also relatively conserved (16 of 23 residues) in $S_9$-SP11, but the amino acid sequence is less conserved (three residues; Supplementary Fig. 4b). These large differences in ligand-receptor contact surfaces between the $S_8$- and $S_9$-complexes enable the $S_8$/$S_9$-discrimination. Conservation analysis revealed that the amino acids constituting the SP11-binding

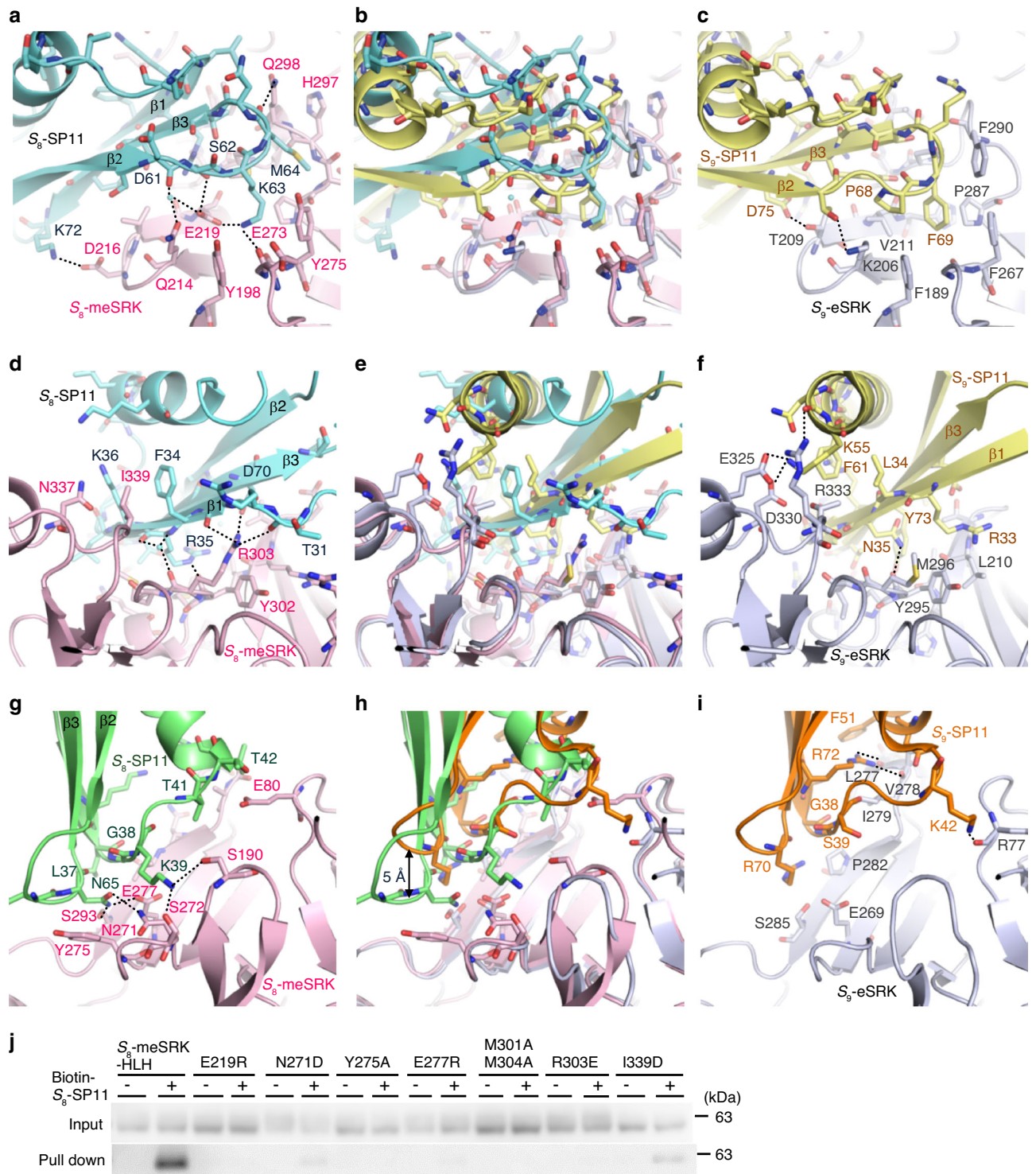

**Fig. 2 Interface between $S_8$-meSRK and $S_8$-SP11. a, d** Close-up views of SP11-binding site 1 on $S_8$-meSRK (pink) with $S_8$-SP11 (cyan). **g** Close-up view of SP11-binding site 2 on $S_8$-meSRK with another molecule of $S_8$-SP11 (green). **b, e, h** Comparison of SP11-binding sites of eSRK between the $S_8$-meSRK–$S_8$-SP11 and $S_9$-eSRK–$S_9$-SP11 complexes. The eSRK molecules were superimposed using $C_\alpha$ atoms and show the same views as in a, d, and g, respectively. $S_9$-eSRK is shown in silver, and $S_9$-SP11 in yellow (interacting with SP11-binding site 1) and orange (site 2). **c, f, i** Close-up views of $S_9$-eSRK–$S_9$-SP11 complex in the same orientation as in (**b**), (**e**), and (**h**). **a–i**, Dotted lines represent hydrogen bonds. Water molecules are shown as small cyan spheres. **j** Pull-down of $S_8$-meSRK-HLH mutants with biotin-$S_8$-SP11 as in Fig. 1b.

pocket are highly variable among the $S$-haplotypes (Supplementary Fig. 3j).

**SRK dimerization in $S_8$- and $S_9$-complexes.** The contact area of the $S_8$-meSRK dimerization surface (916.1 Å$^2$) is smaller than that of $S_9$-eSRK (964.1 Å$^2$; Supplementary Fig. 5a). In the center of the dimerization surface in the $S_8$-meSRK complex, His297, Val313, and Asn314 are in tight contact with the same residues of another $S_8$-meSRK molecule via van der Waals interactions or hydrogen bonds (Supplementary Fig. 5b). By contrast, the

position of the dimerization center in the $S_9$-complex is slightly different because Phe290, which corresponds to His297 in the $S_8$-complex, is in the opposite orientation to His297 (Supplementary Fig. 5c, d). Due to the large distance, Asn307 in $S_9$-eSRK (Asn314 in $S_8$-complex) does not contribute to dimerization, even though the amino acid is conserved (Supplementary Fig. 5c, d). Compared to Ser284 in $S_8$-meSRK, which forms a hydrogen bond with Gln331, Ser276 in $S_9$-eSRK is 7 Å from Ser284 in $S_8$-meSRK in the superimposition of the $S_8$- and $S_9$-complexes and forms a hydrogen bond with Asp330 that contributes to dimerization in the $S_9$-complex, but not the $S_8$-complex (Supplementary Fig. 5e–g). In the 16 residues of $S_8$-meSRK involved in dimerization, only six residues are conserved in $S_9$-eSRK (Supplementary Fig. 5h). These differences in dimerization mode seem to cause a mismatch in the orientation of eSRK between the $S_8$- and $S_9$-complexes (Supplementary Fig. 3b). The variation in dimerization modes may contribute to the suppression of SRK hetero-dimerization in diploid plants.

**Comprehensive analysis of class-I eSRK–SP11 interactions.** Following the success of structure determination, we conducted modeling of the other eSRK–SP11 complexes based on the $S_8$ and $S_9$ crystal structures. Homology modeling of eSRK structures, which belong to class-I haplotypes, was successful because their amino acid sequences are generally quite similar to each other (Supplementary Fig. 6a). To avoid the artificial effects of the $S_8$-meSRK mutations in the following experiments, we also made an $S_8$-eSRK model from $S_8$-meSRK. On the other hand, the homology modeling of SP11 could not produce reliable models because the tertiary structures are expected to vary due to the low sequence homology and differences in sequence length. To overcome this difficulty, we employed accelerated MD (aMD) simulations, which allow access to converged structures on a shorter time scale than conventional MD simulations[31]. Preliminary testing showed that the rmsd value of an $S_9$-SP11 model against the crystal structure generated from a $S_8$-SP11 crystal structure with a conventional homology modeling protocol was reduced to <2.0 Å from the initial value, 3.91 Å, after a 150-ns aMD simulation, demonstrating that the overall tertiary structure was well converged to the crystal structure (Supplementary Fig. 6b–d). Thus, we obtained five reliable SP11 model structures ($S_{32}$, $S_{36}$, $S_{46}$, $S_{47}$, and $S_{61}$) that generally shared the defensin-like domain but exhibited unique structural features (Supplementary Fig. 6e). $S_{46}$-SP11, which has seven Cys residues, formed three disulfide bonds, leaving the first Cys unbonded, whereas the other haplotypes could form four or five disulfide bonds. $S_{32}$- and $S_{36}$-SP11s, which have ten Cys residues, formed an additional disulfide bond relative to general SP11. Using these converged SP11 models, we next constructed docked eSRK–SP11 complex models.

Comprehensive complex structures of self- and nonself-eSRK–SP11 combinations, including the actual $S_8$-eSRK corrected from $S_8$-meSRK, were modeled by superimposing onto the $S_8$ or $S_9$ crystal structure. In the subsequent molecular mechanics–generalized born surface area (MM–GBSA) calculations, the $S_8$-eSRK–$S_8$-SP11 and $S_9$-eSRK–$S_9$-SP11 complexes had binding free energies ($\Delta G$) of –72.48 and –97.97 kcal mol$^{-1}$, respectively, whereas $S_8$-eSRK–$S_9$-SP11 and $S_9$-eSRK–$S_8$-SP11 had energies 30.80 and –5.64 kcal mol$^{-1}$, respectively, indicating that self-combinations form stable complexes, as shown in previous binding experiments[16] (Fig. 3a). Comprehensive analysis of MM–GBSA using twenty eSRK and seven SP11 structures revealed large negative $\Delta G$ values only on eSRK–SP11 complexes of the same haplotype, suggesting that self-pairs, but not nonself-pairs, can form stable complexes as in the $S_8$ and $S_9$ crystal structures.

**Different modes of SP11 recognition in the $S_8$- and $S_9$-subgroups.** Subsequent per-residue energy decomposition analysis of the seven self-eSRK–SP11 complexes revealed that the recognition modes can be categorized into two subgroups, the $S_8$-subgroup ($S_8$, $S_{46}$, $S_{47}$, and $S_{61}$) and $S_9$-subgroup ($S_9$, $S_{32}$, and $S_{36}$), based on the pattern of $\Delta G$ contributions (Fig. 3b, c). The subgroups are consistent with the topology of phylogenetic trees of SRK and SP11, suggesting that the recognition modes are conserved within the subgroups (Supplementary Fig. 7). We also identified three important regions: the C-terminus of the β1 strand, the C-terminus of the α-helix, and the loop structure between β2 and β3 strands in SP11, denoted as Contact Regions I–III (CR I–III), respectively, that predominantly contributed to $\Delta G$ for their corresponding eSRK (Fig. 3b; Table 1). Interestingly, CR I contributed 20.3–30.9% to $\Delta G$ in $S_8$-relative haplotypes, whereas less or no contribution was observed in the $S_9$-subgroup (0.0–10.9%; Table 1). Meanwhile, $S_9$- and $S_{32}$-SP11 had large negative $\Delta G$ values in CR II, consistent with the large eSRK–SP11 interface observed in the $S_9$ crystal structure. CR III, located near the center of the eSRK–SP11 tetramer complex in general, was identified as the most important region for binding in both $S_8$- and $S_9$-relative haplotypes (27.4–55.0%).

Our analysis also identified different binding modes of eSRK between $S_8$- and $S_9$-subgroups (Fig. 3c). A polar interaction between a Glu/Thr residue at position 21 in the eSRK-HV I (e.g., $S_8$-Glu219) and a Lys/Gln at the loop of SP11-CR III ($S_8$-Lys63) is characteristic in the $S_8$-subgroup (Supplementary Fig. 8a, c, d, f, h, j), whereas a hydrophobic interaction is present there in the $S_9$-subgroup (Supplementary Fig. 9). In addition, a Tyr residue at position 7 in the eSRK-HV II (e.g., $S_8$-Tyr275), which is completely conserved in the $S_8$-subgroup, interacts with an Asn residue located on position 55 of SP11 (Fig. 3b), except in $S_{61}$, to contribute to high binding affinity (Supplementary Fig. 8c, f, h, j), although the interaction was absent in the $S_9$-subgroup. Finally, residues on positions 12 and 14 in eSRK-HV III play important roles in the binding mode because they are generally hydrophobic to interact with residues on CR I (mainly positions 8 and 10) in $S_8$-subgroup (Supplementary Fig. 8b, e, g, i), but Asp and Thr residues are present to interact with Lys residue(s) in the C-terminus of CR II instead in $S_9$-subgroup (Supplementary Fig. 9). Taken together, these different interaction modes using CR I–III are largely responsible for global self/nonself-recognition between SP11 subgroups.

**Self/nonself-discrimination in the $S_8$- and $S_9$-subgroups.** Next, we tried to identify residues that cause self/nonself-discrimination within the two subgroups. Because the $S_{46}$-haplotype is closely related to $S_8$ in the $S_8$-subgroup (85% identity in eSRK, 36% in mature SP11), and the $S_{32}$- and $S_{36}$-haplotypes in the $S_9$-subgroup had high sequence identity (88% in eSRK, 75% in mature SP11), we examined the two pairs using computational and experimental analyses. Our $S_{46}$-eSRK–$S_{46}$-SP11 complex model (Fig. 4a) revealed that Ser273 and Asp275 in $S_{46}$-eSRK, which are the residues corresponding to Asn271 and Glu273 in $S_8$-eSRK, respectively, were located close to Asn59 of $S_{46}$-SP11, and that Ile339, which has a more hydrophobic side chain than the same position of $S_8$-eSRK (Asn337), interacted with Phe39 in the α-helix of $S_{46}$-SP11. Hence, we performed an experimental mutation analysis to examine the effect on self/nonself-discrimination. Figure 4b showed that $S_8$-meSRK N271S, E273D, and N337I triple mutations almost completely abolished binding to $S_8$-SP11, although the N271S/E273D double and the N337I single mutants had only limited effects. This observation indicates that the tight binding between the $S_8$-meSRK and $S_8$-SP11 were achieved by multiple residues (or regions). In contrast to the result shown in Fig. 2j, this observation is the consequence of relatively mild

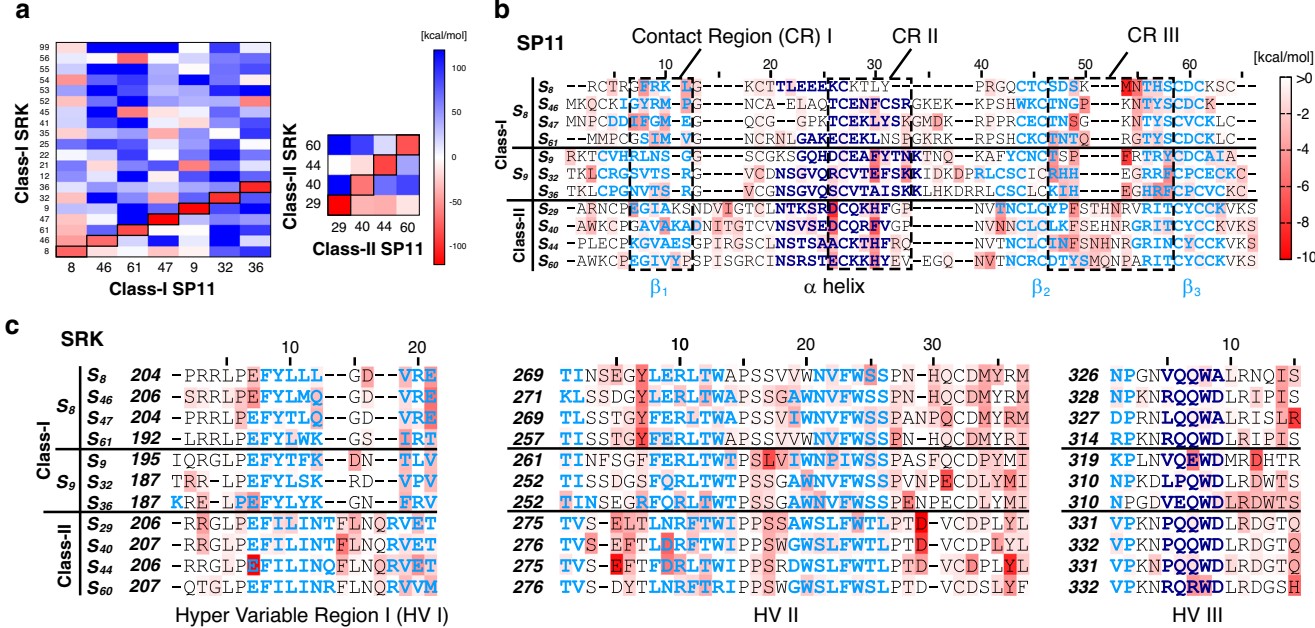

**Fig. 3 MM–GBSA analysis using our modeled eSRK–SP11 complexes. a** Calculated ΔG values for modeled SP11s against eSRKs. Red rectangles represent tightly bound eSRK–SP11 pairs, and white to blue rectangles represent pairs with weak affinity. These values were calculated by the MM–GBSA method. **b**, **c** Sequence alignments of SP11 (**b**) and the hypervariable regions (HVs) of SRK (**c**) haplotypes with heat maps indicating energy contributions from each residue. Residues consisting of an α-helix and β-strands are shown in dark and light blue, respectively. These alignments were generated using PROMALS3D[70].

---

## Table 1 Proportion of ΔG contribution from SP11.

| | CR I[a] | CR II[a] | CR III[a] |
|---|---|---|---|
| $S_8$-subgroup (class-I) | | | |
| $S_8$-SP11 | 22.2% | 0.4% | 50.0% |
| $S_{46}$-SP11 | 20.5% | 11.0% | 38.7% |
| $S_{47}$-SP11 | 30.9% | 14.5% | 27.4% |
| $S_{61}$-SP11 | 20.3% | 6.8% | 48.8% |
| $S_9$-subgroup (class-I) | | | |
| $S_9$-SP11 | 0.0% | 23.5% | 47.2% |
| $S_{32}$-SP11 | 1.5% | 27.4% | 29.8% |
| $S_{36}$-SP11 | 10.9% | 7.0% | 55.0% |
| Class-II subgroup | | | |
| $S_{29}$-SP11 | 11.2% | 29.1% | 20.1% |
| $S_{40}$-SP11 | 8.4% | 22.0% | 14.1% |
| $S_{44}$-SP11 | 3.0% | 24.7% | 45.8% |
| $S_{60}$-SP11 | 11.9% | 24.7% | 23.5% |

[a]CR I, II, and III correspond to the C-terminal β1 strand (Position 7–12), C-terminal α-helix (26–33), and β2–β3 loop (47–58), respectively (see also Fig. 3b).

---

differences in amino acid characteristics between $S_8$-SRK and $S_{46}$-SRK. For $S_{32}$- and $S_{36}$-haplotypes, on the other hand, we identified Ser36, Lys57, Ile58, and His62 residues in $S_{36}$-SP11 as candidate residues involved in discrimination between $S_{32}$- and $S_{36}$-eSRK based on our models (Fig. 4c). To confirm the importance of the residues, we performed a pollination bioassay[16]. When an $S_{36}S_{36}$ pistil was treated with recombinant $S_{36}$-SP11 protein, compatible $S_{12}$ pollen was rejected by $S_{36}$-SP11-induced SI reaction. Among the residues, only the $S_{36}$-SP11[H62R] mutant did not induce the SI reaction against $S_{36}S_{36}$ pistils, suggesting that the mutation critically disrupted the formation of the SRK–SP11 complex (Fig. 4d). This observation can be explained by our $S_{36}$ complex models because $S_{36}$-eSRK Arg258, which is anomalously

located at this position but not in any other eSRK structure, can interfere with H62R mutation through steric and electrostatic repulsion (Fig. 4c).

**Computational analysis in class-II haplotypes.** To show that our computational methodology can be applied to other subgroups, we also modeled eSRK and SP11 structures belonging to class-II haplotypes, $S_{29}$, $S_{40}$, $S_{44}$, and $S_{60}$ (hereafter the class-II subgroup). Class-II haplotypes, which exhibit a recessive SI phenotype in the pollen part against other class-I haplotypes, including the $S_8$- and $S_9$-subgroups, are phylogenetically farthest from known class-I haplotypes in *B. rapa* (Supplementary Fig. 7)[32–36]. Although sequence homology between the class-II SRK haplotypes is high, there is a characteristic four-residue amino acid insertion in the HV I region relative to the $S_8$- or $S_9$-subgroup. We performed homology modeling for all four class-II eSRK proteins, using the $S_8$-meSRK crystal structure as a template with the four-residue insertion (FLNQ) set to a β-turn structure, and successfully obtained stable model structures. On the other hand, because of low homology with $S_8$- or $S_9$-SP11 sequences and the absence of analogous structures in PDB, the Rosetta ab initio protocol[37] (see "Methods") was used to generate initial structures of class-II SP11 proteins. The backbones of final class-II SP11 models, after refinement with aMD simulations, are similar to those of other class-I SP11 proteins (Supplementary Fig 6e).

After superimposing the modeled class-II eSRK and SP11 structures on the $S_8$ crystal structure, we carried out extensive aMD simulations to search for the binding positions of SRK and SP11. Similar to the results of the $S_8$- and $S_9$-subgroups, MM–GBSA calculations showed that the class-II eSRK proteins could bind to their corresponding SP11 proteins strongly, but not otherwise (Fig. 3a). All four class-II eSRK–SP11 complex models obtained from the aMD simulations revealed that a Phe/Tyr residue in the β2 strand of SP11-CR III (e.g., Phe75 in $S_{29}$) interacted with a Phe residue (e.g., Phe218 in $S_{29}$) located in the

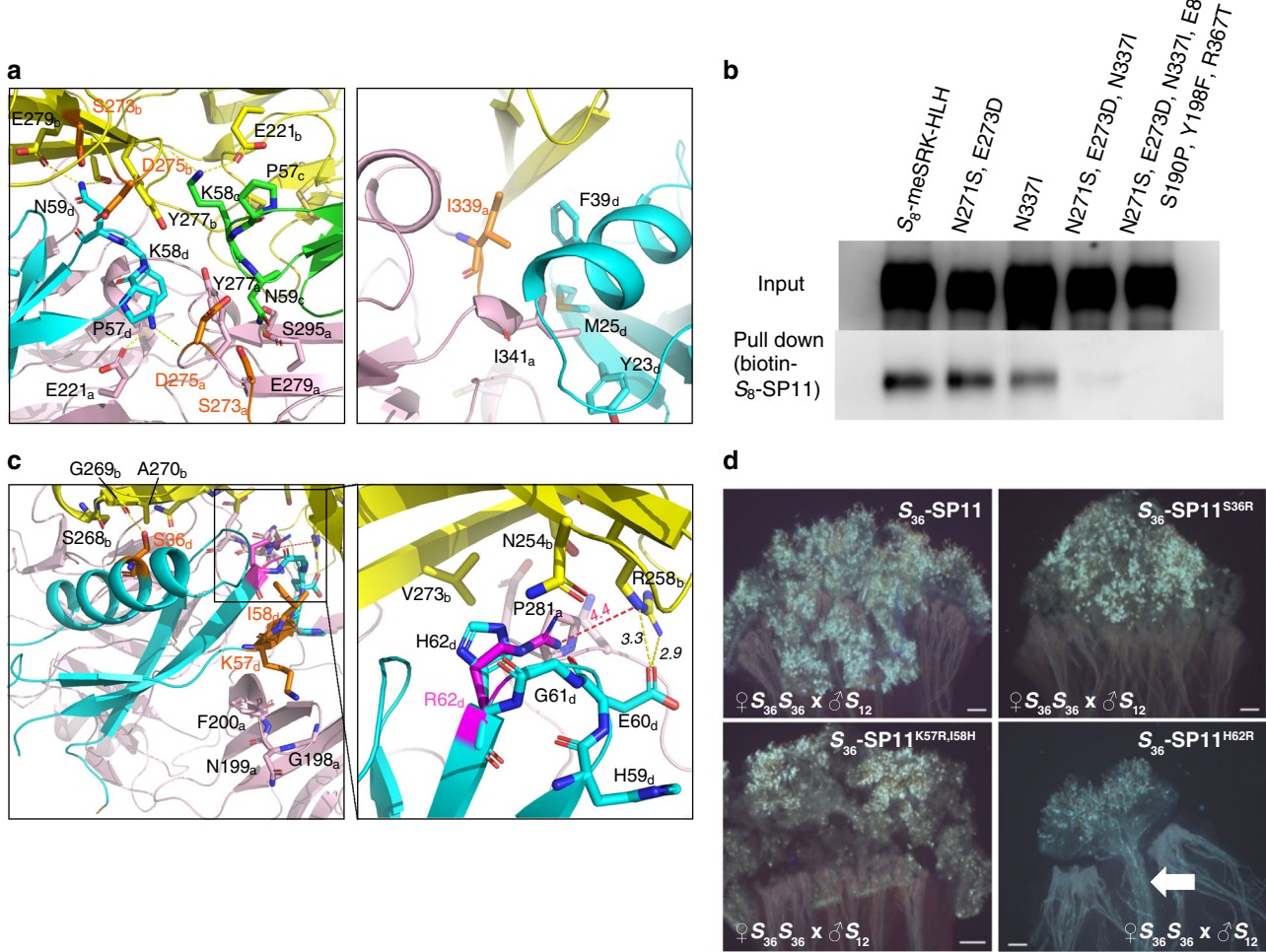

**Fig. 4 Identifying important residues for self/nonself-discrimination between similar eSRK–SP11 pairs. a** Model structure of $S_{46}$-eSRK–$S_{46}$-SP11 complex. Green and cyan represent two $S_{46}$-SP11 molecules in the heterotetramer, and pink and yellow represent two $S_{46}$-eSRK molecules. Residues mutated in the pull-down assay are shown in orange. Left and right panels show close-up views of interfaces around CR III and around CR I–II, respectively. **a**, **c** Subscripts to the right of residue numbers indicate chain ID in the complex (**a**, **b**, eSRK; **c**, **d**, SP11). **b** Pull-down of $S_8$-meSRK-HLH and its derivatives with biotin-$S_8$-SP11 was performed as in Fig. 1b. $S_8$-meSRK-HLH$^{N271S,E273D,N337I}$ and $S_8$-meSRK-HLH$^{N271S,E273D,N337I,E80G,S190P,Y198F,R367T}$ proteins lost the ability to bind $S_8$-SP11. **c** Model structure of the $S_{36}$-eSRK–$S_{36}$-SP11 complex. An overview of $S_{36}$-SP11 docked with $S_{36}$-eSRK dimer is shown in the left panel. Cyan represents $S_{36}$-SP11 molecule, and pink and yellow represent two $S_{36}$-eSRK molecules. Residues mutated in the bioassay are shown in orange. A close-up view of the CR III of $S_{36}$-SP11 is shown in the right panel. The H62R mutation is shown in pink. Distances are shown in angstroms. **d** Pollination bioassay. $S_{36}$-SP11- or mutant-treated (50 pmol) $S_{36}S_{36}$ pistils were pollinated with $S_{12}$ pollen grains. Pollen tubes were observed by UV fluorescence microscopy after aniline blue staining. Arrow shows pollen tubes. Scale bars, 100 μm.

four-residue insertion in eSRK-HV I (Fig. 3b, c; Supplementary Fig. 10). In addition to this aromatic–aromatic interaction, SP11-His/Phe62 in CR II, which is present in the C-terminus of the α-helix, could enter a cleft between eSRK-HV I and HV III, and its main chain formed a hydrogen bond with a Lys residue of eSRK (e.g., K333 in $S_{29}$). These observations indicated that these interactions characterize the binding mode of class-II eSRK–SP11 complexes. Moreover, the per-residue energy decomposition analysis (Table 1) showed that the contribution of CR II to ΔG in the class-II subgroup was relatively high (22.0–29.1%) whereas that of CR III was slightly lower (14.1–45.8%) than in the $S_8$- or $S_9$-subgroup, also suggesting a difference in their binding modes.

These eSRK–SP11 complex models and MM-GBSA calculations also provide insight into self/nonself-discrimination within class-II haplotypes. Although the amino acid sequences of the HV regions of SRK are relatively conserved in the four class-II haplotypes, the residues that formed an interface between the β-turn structure of eSRK-HV II and the α-helix of SP11 were

completely different, suggesting that these residues can generate self/nonself-discrimination. For example, the 58th residue of SP11 (position 26 in Fig. 3b) is Ala in $S_{44}$, but Asp or Glu in the other three haplotypes. The residue is proximal to the 292nd residue of eSRK-HV II (position 19 in Fig. 3c), which is Asp in $S_{44}$ but Gly or Ala in the others (Supplementary Fig. 10). If these negatively charged amino acids are present in both eSRK and SP11, it may impair the formation of the eSRK–SP11 complex and contribute to self/nonself-discrimination within class-II haplotypes. Moreover, Arg82 of $S_{40}$-eSRK may contribute to discrimination between $S_{29}$- and $S_{40}$-SP11 because the bulkier and positively charged Lys55 of $S_{29}$-SP11 can interfere with it, whereas Val55 of $S_{40}$-SP11 cannot (Supplementary Fig. 10a, b). In addition, the presence of additional mutations in $S_{60}$ may contribute to discrimination from the other class-II haplotypes. Arg338 of $S_{60}$-eSRK, located on HV III, could form a salt bridge with Glu58 of $S_{60}$-SP11. It should also be noted that Tyr192 of $S_{60}$-eSRK, which is not present in the HV regions but is unique to $S_{60}$, is located

near Met80 of $S_{60}$-SP11, forming a methionine-aromatic interaction[38] (Supplementary Fig. 10d). These additional interactions contributed to large negative $\Delta G$ in the $S_{60}$ complex, as shown in Fig. 3. Taken together, our computational models for the class-II subgroup revealed that the binding mode between eSRK and SP11 was significantly different from that of the $S_8$- or $S_9$-subgroup, and that the self/nonself-discrimination within the class-II subgroup would be achieved by amino acid variation in the shared binding interfaces.

## Discussion

Our computational modeling using the two crystal structures successfully generated highly reliable models of the eSRK–SP11 complex and identified a global difference in eSRK–SP11 recognition among the subgroups in the MM–GBSA and local interactions that discriminate between similar haplotypes. Phylogenetic analysis suggested that the *SRK* and *SP11* genes in *Brassica* have co-evolved to maintain stable interactions between self-combinations (Supplementary Fig. 7)[27,39,40]. Our data suggest that the S-haplotypes can be classified into a small number of subgroups with similar recognition modes based on the topology of the phylogenetic trees (Supplementary Fig. 7). Some key residues important for self-recognition within each subgroup appear to be evolutionarily restricted, because mutations in these residues are more likely than changes in minor residues to abolish recognition ability, as in the case of other highly conserved amino acids required for protein function. Therefore, the recently emerged S-haplotypes are likely to have been primarily generated by mutations in minor residues.

The haplotype-specific interaction has been explained based on amino acid variation in the binding surfaces of SRK and SP11[29]. However, our research suggests that the distinctions between multiple contact regions (HV/CR) in both SRK and SP11 are also important for the haplotype-specific interaction. The frequency of HV/CR usage, as well as amino acid variation, enables robust and rigorous recognition of self-SP11, and also seems to have aided the acquisition of numerous S-haplotypes over the course of evolution by ensuring the multiplicity of SRK–SP11 interaction surfaces. We demonstrated that our methods can be used to analyze the interactions between SRK and SP11 in other unrelated subgroups and have potential applications for future analysis to identify unknown pairs of defensin-like ligands and SRK-like receptors. Our results contribute to the comprehensive understanding of not only self/nonself-discrimination in *Brassica* SI, but also other related protein–protein interactions.

## Methods

**Constructs.** SRK[28,30,32–34,39] and SP11[9–11,27,35] sequences used in this study are listed in Supplementary Data 1. For protein expression, $S_8$-eSRK (encoding residues 32–433) was cloned into pBac6 (Merck Millipore) as an N-terminal 6xHis and C-terminal DYKDDDDK (FLAG)-tag fusion protein connected by the Human Rhinovirus 3C (HRV3C) protease recognition linker. $S_8$-meSRK, which encodes $S_8$-eSRK containing 11 amino acid mutations (P79S, Y80E, I81R, F108V, L110R, L180R, F190S, L239S, L214Q, V286G, and V287A), was synthesized by step-by-step site-directed mutagenesis using the KOD -Plus- Mutagenesis Kit (TOYOBO). To suppress recombinant protein aggregation, the 11 amino acids were replaced with less hydrophobic residues of $S_8$-SLG or other SRK proteins. The P79S, Y80E, and I81R mutations were derived from the $S_8$-SLG sequence; F108V and L110R are from $S_{12}$-SRK; L180R is from $S_{60}$-SRK; F190S and L239S are from $S_9$-SRK; and L214Q, V286G, and V287A are from $S_{46}$-SRK. $S_8$-meSRK-HLH encodes $S_8$-meSRK with a C-terminal fusion of the SREBP-2 dimerization domain (HLH, residues 343–403), as described previously[18,41] and the same tags were cloned into pBac6. $S_8$-SP11 (encoding residues 25–74) and $S_{36}$-SP11 (14–71) were cloned into pET39b (Novagen) as DsbA fusions with the HRV3C protease recognition linker. Mutations were introduced into $S_8$-meSRK, $S_8$-meSRK-HLH, and $S_{36}$-SP11 constructs by site-directed mutagenesis.

**Structure determination.** $S_8$-eSRK and the derivatives were expressed as secreted proteins in an insect cell system. Baculovirus incorporating each eSRK construct

was generated using the BacMagic system (Merck Millipore). For $S_8$-meSRK crystallization, 10 mL of P4 virus was used to infect 1 L of Sf9 cell culture ($1.5 \times 10^6$ cells mL$^{-1}$) grown in Sf-900 II SFM (Thermo Fisher) medium containing 1% FBS, 100 µg mL$^{-1}$ streptomycin, and 0.25 µg mL$^{-1}$ amphotericin B, and cultured for 72 h at 23 °C. The culture medium was concentrated using a VIVAFLOW 200 (Sartorius) and subjected to DYKDDDDK antibody–conjugated agarose resin column (Wako). After washing with Buffer A (20 mM Tris-HCl, pH 8.0 and 100 mM NaCl), $S_8$-meSRK was eluted with Buffer A containing 0.3 mg mL$^{-1}$ DYKDDDDK peptide and 200 mM arginine. Tags were removed by 12-hour incubation with HRV3C protease at 4 °C. Further ion-exchange (HiTrap Q, GE Healthcare) and gel-filtration (Superdex 200, GE Healthcare) chromatography yielded a single protein band in SDS-PAGE analysis. A total of 5 mg $S_8$-meSRK was obtained from 250 L culture and concentrated to 7–9 mg mL$^{-1}$ for crystallization. Purified protein was confirmed by MALDI-TOF-MS and protein sequence analysis. Selenomethionine (SeMet)-labeled $S_8$-meSRK was expressed in ESF 921 methionine-deficient medium (Expression Systems) supplemented with 100 mg L$^{-1}$ SeMet at 24 and 48 h after baculovirus infection. Other culture conditions and protein purification steps are same as for the native protein.

$S_8$-meSRK and chemically synthesized biotin-$S_8$-SP11[16] were mixed at a 1:3 molar ratio. $S_8$-meSRK–$S_8$-SP11 complex was crystallized by the vapor diffusion method in a mixture of 1 µl of protein solution and 1 µl of reservoir solution containing 15–16% PEG3350 and 0.2 M magnesium formate. Crystals were transferred into cryoprotectant solution containing 30% PEG3350, 0.2 M magnesium formate, and 5% glycerol (native crystal) or 16% PEG3350, 25% PEG200, and 0.2 M magnesium formate (SeMet-labeled) and flash-frozen in liquid nitrogen. Data collections were performed at beamlines BL41XU and BL44XU at SPring-8 and beamline BL1A at the Photon Factory. Data collection of SeMet-labeled crystals was performed at a peak wavelength of Se (0.9791 Å). Diffraction data were processed and scaled using HKL2000[42], and statistics of data collection and processing are summarized in Supplementary Table 1. We cut off the native data at 2.6 Å resolution due to the high $R_{merge}$ value (0.8>), even though the CC1/2 and $I/\sigma I$ values were still enough.

The initial phase was determined by SAD (single-wavelength anomalous dispersion) approach. Eleven Se atoms were found by SHELX[43] program using 3.5 Å diffraction data from SeMet-labeled $S_8$-meSRK–$S_8$-SP11 crystal. The initial phase was solved by AutoSol in the Phenix program suite[44], and model building was performed using AutoBuild in Phenix. The initial model was used for molecular replacement by Phaser[45] using 2.6-Å diffraction data from the native crystal. The model was refined using phenix.refine in Phenix and COOT[46], and validated with MolProbity[47]. Refinement statistics are shown in Supplementary Table 1. Protein–protein interaction area was calculated using areaimol in CCP4 suit[48]. Structural figures were created using PyMol (https://pymol.org/2/) with the APBS plugin. The conservation profile was generated using the ConSurf server (https://consurf.tau.ac.il/).

**Isothermal titration calorimetry (ITC).** ITC measurement of the $S_8$-meSRK–$S_8$-SP11 interaction was carried out on a MicroCal iTC200 (Malvern Panalytical) at 20 °C. Protein solutions were prepared at 29 µM ($S_8$-meSRK) and 290 µM (biotin-$S_8$-SP11) in Buffer A. For one titration cycle, 1 µl of $S_8$-SP11 solution was injected into a sample cell containing 200 µL of $S_8$-meSRK. The run consisted of 39 cycles and 5-minute intervals. The data were analyzed using the ORIGIN software (Malvern Panalytical).

**Gel-filtration analysis.** Gel-filtration analysis of $S_8$-meSRK–$S_8$-SP11 interaction was performed on a Superdex 200 10/30 column (GE Healthcare). $S_8$-meSRK (10 µM), $S_8$-SP11 (30 µM), and the mixture in Buffer A were independently analyzed at 4 °C. Molecular marker (Bio-Rad) was also used for molecular size estimation. Full image of SDS-PAGE gel is shown in Supplementary Fig. 11.

**NMR analysis.** One milligram of $S_8$-meSRK protein was prepared from 50 L culture of Sf9 cells as described above. $^{15}$N- and $^{15}$N/$^{13}$C-labeled $S_8$-SP11 were expressed as DsbA fusion proteins in *Escherichia coli* Rosetta 2 (DE3) strain (Merck Millipore) in M9 medium containing 0.5 g L$^{-1}$ $^{15}$N-ammonium chloride (Cambridge Isotope Laboratories) and 1 g L$^{-1}$ $^{13}$C-glucose (Cambridge Isotope Laboratories) as the sole sources of nitrogen and carbon, respectively. The recombinant protein was purified from cell lysate by nickel column and ion-exchange (HiTrapQ) chromatography. After digestion with HRV3C protease, $^{15}$N- and $^{15}$N/$^{13}$C-labeled $S_8$-SP11 were further purified using Superdex200 and ODS (Vydac) columns. To establish the NMR assignments, a series of 3D NMR experiments (HNCACB, HNCA, CBCA(CO)NH, HN(CA)CO, and HNCO) were performed. NMR experiments were performed at 298 K using a Bruker AVANCEIII 600 equipped with a TCI CRYOPROBE and a Bruker AVANCEIII HD 900 equipped with a TCI CRYOPROBE. NMR spectra were processed and analyzed using NMRPipe[49] and Sparky (Goddard, T. D. & Kneller, G. D.; https://www.cgl.ucsf.edu/home/sparky/), respectively.

**Pull-down assay.** All pull-down assays were performed using $S_8$-meSRK or $S_8$-meSRK-HLH proteins. For the mutational analysis, the E219R, N271D, Y275A, E277R, M301A/M304A, R303E, and I339D mutations in $S_8$-meSRK-HLH were tested. For $S_8$-meSRK expression, baculovirus encoding each gene was used to infect Sf9

cells, and the infected cells were cultured at 23 °C for 72 h. The supernatants (1 mL) were mixed with 1 μg of biotin-$S_8$-SP11 and incubated at 4 °C for 2 h. After the addition of 30 μL of avidin beads (PIERCE), the samples were gently agitated at 4 °C for 2 h. The avidin beads were washed five times with Buffer A, and binding proteins were eluted by SDS sample buffer. The samples were separated by SDS-PAGE and transferred to PVDF membranes (Merck Millipore). Immunoblots were performed using SNAP i.d. (Merck Millipore); Blocking One (Nacalai Tesque) was used for blocking, Can Get Signal (TOYOBO) for as solvent, and TTBS (10 mM Tris-HCl [pH7.4], 100 mM NaCl, and 0.05% Tween 20) as washing buffer. Primary antibody C1[18], which recognizes $S_8$-SRK and $S_8$-SLG, and secondary antibody anti-rabbit HRP (Bio-Rad) were used at a dilution of 1:4,000. The signals were detected on a LAS4000 (Fujifilm) using Luminata Forte Western HRP Substrate (Merck Millipore). The full images of blots are shown in Supplementary Fig. 11.

**Pollination bioassay.** For the stigma preparation, non-pollinated mature flower buds were cut at the peduncle, and the anthers were cut off and stood on 1% solid agar plate. For the pollination bioassay, stigmas from $S_{36}$ homozygous plants were treated with peptides containing 0.05% Tween 20, and then dried in air for 1 h. After pollination with the pollen from an $S_{12}$ plant, the stigmas were kept at 23 °C overnight. Thereafter, pistils of pollinated flowers were softened with 1 N NaOH for 1 h at 60 °C, and then stained with basic aniline blue (100 mM $K_3PO_4$, 0.1% aniline blue). Samples were mounted on slides in 50% glycerol (fluorescence microscope grade) and observed by UV-fluorescence microscopy[50].

**Computational methods.** *Modeling of SP11*: Tertiary structures of SP11 proteins were generated by a two-step process: initial structure modeling and aMD simulations. First, sequences of $S_{46}$, $S_{47}$, $S_{61}$, $S_{32}$, and $S_{36}$-SP11 proteins were aligned using ClustalW version 2.1[51,52] so that all cysteine residues were located at the same positions. Coordinates of $S_8$-SP11 taken from the crystal structure (PDB ID: 6KYW) were used as a template for haplotypes $S_{46}$, $S_{47}$, and $S_{61}$, whereas $S_9$-SP11 taken from $S_9$-eSRK–$S_9$-SP11 crystal structure (PDB: 5GYY)[29] was used for $S_{32}$ and $S_{36}$. Five homology models were generated using MODELLER version 9.16[53,54], and the one with minimal 'molpdf' score was selected as the initial structure for subsequent MD simulations. Sequences used for the modeling are described in Supplementary Data 1.

For modeling of the class-II SP11 proteins ($S_{29}$, $S_{40}$, $S_{44}$, and $S_{60}$), the initial structure was generated by ab initio modeling using Rosetta 3.9[37] instead of homology modeling because of their sequences were very divergent from those of class-I SP11. We assumed that all eight cysteines at the C-terminus of the class-II SP11 proteins could also form disulfide bonds to form a defensin-like domain, as observed in $S_8$- or $S_9$-SP11, and searched for the combinations of the length of secondary structures and disulfide pairs. The best one is shown in Supplementary Fig. 6e. We selected a model structure with the minimal Rosetta score as the initial structure for subsequent MD simulations.

After adding hydrogen atoms, the modeled SP11 structure was fully solvated in the TIP3P water model[55] in a cubic periodic box, and then neutralized by adding $Na^+$ and $Cl^-$ ions via the Amber LEaP module[56]. The ff14SB force field[57] was used for the protein. Short-range van der Waals and electrostatic interactions were cut off beyond 10 Å, and the particle mesh Ewald (PME) method[58,59] was used for long-range interactions. The system was first relaxed using 200 steps of the steepest descent minimization, with a 1000 kcal mol$^{-1}$ Å$^{-2}$ constraint applied to the heavy atoms of the protein. Subsequently, the entire system was subjected to 200 steps of the steepest descent minimization without restraints. Next, to gradually heat the system, 1-ns MD simulations were performed at 300 K and $1.0 \times 10^5$ Pa under *NPT* ensemble. During the equilibrations, the SHAKE algorithm[60] was used to constrain the bonds including hydrogen atoms, and the integration time step was set to 2 fs. The Berendsen weak coupling algorithm[61] was used to maintain constant temperature and pressure. After equilibration, 20-ns conventional production runs were carried out. Equilibrations and production runs were performed using the PMEMD module of AMBER 18[62].

To enhance the conformational sampling space and obtain the converged structures of modeled SP11s, 150-ns aMD simulations were performed after the conventional MD simulations described above. In this study, we employed the "dual-boost" aMD, in which a non-negative dihedral boost potential was applied to all dihedrals in the system, in addition to a total boost potential to all atoms in the system. Reference energies for dihedrals ($E_{dihed}$), total potentials ($E_{total}$), and the acceleration factors for them ($\alpha_{dihed}$ and $\alpha_{total}$) were determined as follows:

$$E_{dihed} = 4.0 N_{res} + V_{dihed\_avg}, \quad \alpha_{dihed} = 4.0 N_{res} \times 0.2$$
$$E_{total} = 0.2 N_{atoms} + V_{total\_avg}, \quad \alpha_{dihed} = 0.2 N_{atoms},$$

where $N_{res}$ is the number of protein residues, $N_{atoms}$ is the total number of atoms in the system, and $V_{dihed\_avg}$ and $V_{total\_avg}$ are the average dihedral and total potential energies calculated by the previous 20-ns conventional MD simulations, respectively.

*MD simulations of complex model of eSRK/SP11*: Structures of class-I ($S_{46}$, $S_{47}$, $S_{61}$, $S_{32}$, and $S_{36}$) and class-II ($S_{29}$, $S_{40}$, $S_{44}$, and $S_{60}$) eSRK proteins were modeled in the same manner as homology modeling of SP11 proteins. For the class-II eSRK proteins, the additional four residues located in the HV I region were assigned to form a β-turn structure. To build an initial structure of eSRK–SP11 complex, the

same haplotypes of the modeled eSRK and SP11 were superposed onto the crystal structure of $S_9$-eSRK–$S_9$-SP11 structure for $S_{32}$ and $S_{36}$, $S_8$-meSRK–$S_8$-SP11 for $S_{46}$, $S_{47}$, $S_{61}$, and four class-II haplotypes. Energy minimization and heating of the system for the modeled eSRK–SP11 were then conducted as described above. Unrestrained 150-ns production runs were subsequently carried out at constant temperature (300 K) using the *V*-rescale algorithm[63–65].

*MM-GBSA*: The MM–GBSA[66] implemented in AmberTools 18 was employed to calculate the binding free energy, $\Delta G_{bind}$, for complexes between the modeled eSRK and SP11 structures. The GBOBC implicit solvent model (parameters $\alpha = 1.0$, $\beta = 0.8$, and $\gamma = 4.85$)[67] was used with a salt concentration of 0.2 M. All calculations were performed using MD trajectories between 30 and 150 ns, recorded every 100 ps (1200 snapshots for each complex).

**Evolutionary analysis.** The evolutionary history of *SP11* and *SRK* was inferred using the Neighbor-Joining method[68]. The percentages of replicate trees in which the associated taxa clustered together in the bootstrap test (1000 replicates) are shown next to the branches. The evolutionary distances, in units of number of base substitutions per site, were computed using the Kimura two-parameter method implemented in MEGA X[69].

**Reporting summary.** Further information on research design is available in the Nature Research Reporting Summary linked to this article.

## Data availability

Structure factor and coordinates of the $S_8$-meSRK–$S_8$-SP11 complex have been deposited in PDBJ (Protein Data Bank Japan) with accession number 6KYW [PDB]. Other data are available from the corresponding authors upon reasonable request. Source data are provided with this paper.

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

## Acknowledgements

We thank K. Maenaka for critical advice on protein expression, SPring-8, and Photon Factory (PF) staff for assistance of data collection, and R. Kurata and H. Okubo for MALDI-TOF-MS and protein sequence analysis. This work was supported by JSPS KAKENHI grant numbers 15K18683 and 20K05824 (to K.M.), 18KT0048 and 19K22342 (to M.W.), and 16H06380 (to S.T.); JSPS Bilateral Programs grant number 18032211-000481 (to M.W.); MEXT KAKENHI grant numbers 23113006 (to G.S.), 16H06470, 16H06464, and 16K21727 (to M.W.), and 16H06467 (to S.T.); the Platform Project for Supporting Drug Discovery and Life Science Research (Basis for Supporting Innovative Drug Discovery and Life Science Research [BINDS]) from AMED under Grant Number JP20am0101107 (support number 0840); The Nakajima Foundation (to K.M.); the Takeda Science Foundation (to K.M.); and the Toray Science Foundation (to K.M.). K.M.

was supported by a Postdoctoral Fellowship for Young Scientists from JSPS. Data collection experiments at SPring-8 and PF were supported by JASRI (Proposal numbers: 2015A6549, 2015B6549, 2016A2519, 2016B2519, 2017A6759, 2017B6759, 2018A2529, 2018B2529, and 2019B2516) and KEK (Proposal number, 2014G706). The NMR experiments were partly performed at RIKEN of the NMR Platform supported by the MEXT.

## Author contributions

K.M., T.H., and S.T. conceived the project; K.M., T.M., X.L., C.M., R.M., M.M., S.F., Y.H., Z.K., and K.N. performed structure determination and biochemical experiments; Y.M., T.T., and K.S. designed and performed MD simulation experiments; Y.T., G.S., and M.W. performed the cDNA cloning and bioassay; K.M., Y.M., K.S., T.H., and S.T. wrote the manuscript, which was edited by all other authors.

## Competing interests

The authors declare no competing interests.
