## [Peer Review File · Nature Communications]

REVIEWER COMMENTS

Reviewer #1 (Remarks to the Author):

Self-incompatibility (SI) process is mainly controlled by the interaction between SRK at plasma membrane of papilla at pistil and the SP11 pollen secreted protein. SRK is a receptor Kinase that consists of an extracellular domain, a TM segment and an intracellular Kinase domain. SP11 is a small disulfide rich protein. The extracellular domain of SRK recognizes SP11 and promotes self-phosphorylation of the kinase domain. Both proteins are highly polymorphic. SRK protein contains three hypervariable regions and SP11 variants display few structural similarities, being restricted to the disulfide bond pattern. The specific interaction between different isoforms controls pollen rejection in SI.

The authors determine the crystal structure of the complex between S8-SRK-S8-SP11 from brassica and compare this structure with that of S9-SRK-S9-SP11 from *B. rapa*. They hypothesize that the observed differences are the determinants for self and non self-discrimination for SI. To test this hypothesis, the authors conduct the modeling of other SRK-SP11 complexes based on the reported structures. The difficult part relies on the SP11 moiety as different isoforms display low sequence similarity and length. Despite this difficulty, they build five different models for SP11 isoforms. Then, they construct SRK-SP11 models and calculate the corresponding binding free energies. The data corroborate that only the complexes between pairs of the same haplotype form stable complexes. Finally, the authors validate their model by site directed mutagenesis followed by pull down assays and pollination bioassays.

The manuscript is well written, the introduction is informative, the experimental and theoretical approaches are sound and the methods section is complete. The authors present an interesting crystallographic work dealing with a very difficult problem. However, the figures could be improved to complement the text and the manuscript may benefit from a revision at several levels.

The differences on the overall structures (lines 91 to 105) and dimerization interface (lines 145 to 163) of S8-SRK-S8-SP11 and S9-SRK-S9-SP11 provide the structural basis for the self-discrimination ability. Consequently, they deserve a panel in figure 2. Conversely, the authors provide a very detailed description on the binding sites 1 and 2 that it is difficult to follow; this description needs to be more concise. In this direction, Figure 2 should be changed to compare side by side the details of S8-SRK-S8-SP11 and S9-SRK-S9-SP11 binding sites in the same orientation. The superimpositions shown in panels 2b,d and f confuse the reader.

The theoretical calculations validate well the conclusions drawn from the joined analysis of the crystal structures of S8-SRK-S8-SP11 and S9-SRK-S9-SP11. They provide elegant atomistic models and energy calculations that only account for the already predicted self/nonself-combinations. Indeed, this was pointed out in the paper by Ma et al. 2016. This should be made clear in the discussion section.

Other comments

The authors include 11 point mutations in the extracellular domain SRK to obtain functional protein for structural studies. The methods section would benefit from a description of the procedure that inspired these mutations.

The I/sI value in the highest resolution shell (5.38., supp. Table 1) suggests the diffraction data goes well beyond the 2.6 Å cutoff applied by the authors. Please provide an explanation in the methods section. Besides, the values of the CC1/2 and the R/Rfree in the highest resolution shell should be also provided.

Line 188 "MM-GBSA (Molecular Mechanics-Generalized Born Surface Area)" should be written "Molecular Mechanics-Generalized Born Surface Area (MM-GBSA)"

Reviewer #2 (Remarks to the Author):

The manuscript by Murase et al., 'Mechanism of self/nonself-discrimination in Brassica self-incompatibility' describes molecular dynamics simulations performed based on the S8-SRK-S8-SP11 crystal structure. Based on this, they predict that the binding free energies are most stable between the haplotype-specific combinations. The crystal structure of the receptor and ligand complex revealed 31 contact sites in SRK8 interacting with 23 residues of S8-SP11. Although most of the interacting positions are similar in the SRK9/S9-SP11 complex, there is poor conservation of the interacting residues. In addition, comparison of homodimers of SRK8 and SRK9 revealed a poor conservation of residues that are involved in homodimerization suggesting that this is a means to suppress heterodimerization from occurring between these receptors. Using the crystal structures MM-GBSA calculations were performed between 20 SRKs and 7 SP11 combinations which showed lowest binding free energies from cognate pairs suggesting that only self-pairs can form stable complexes. The authors then were able to model intragroup and intergroup interactions and identify residues that are most important for the interacting cognate pairs.

Solving the S8-SRK-S8-SP11 in addition to the existing S9-SRK-S9-SP11 structure has allowed the authors to come up with the various models and interacting residues that determine the specificity of interaction between self-pairs. This is certainly a significant advancement in the field. The modeling allows convenient and precise restructuring of the receptor or the ligand in order to either force an interaction or uncouple an interaction.

I have some minor concerns about the story:

Did any of the 11 mutations to make the SRK8 protein stable correspond to the hypervariable regions I and II of SRK? Are these mutated sites conserved across haplotypes and whether this can influence inter and intra subgroup combination. Were any of these mutations located in the 31 contact position of SRK8 with S8-SP11? Authors need to explain the influence of these mutations on binding other SP11 haplotypes. Highlight the mutated residues in Supplementary figure 4.

The authors claim that multiple residues in SRK8 are required for the tight binding of SRK8 with S8-SP11 (Lines 239 to 243, Fig. 4b). Contrarily in Fig. 2g, multiple single mutants are shown to abolish SRK8 binding to S8-SP11 including N271D mutation. Please address this discrepancy in the text.

Reviewer #3 (Remarks to the Author):

Self-incompatibility (SI) in Brassica is mediated by specific interaction between the pistil SI determinant SRK and the pollen SI determinant SP11. SP11 of a given S-haplotype can only be recognized by the SRK of the same S-haplotype, and the haplotype-specific interaction results in downstream signaling events, leading to rejection of self-pollen. In order to understand the biochemical basis of self-recognition between SP11 and its cognate SRK, the authors, in the work reported in this manuscript, determined the crystal structure of the extracellular domain of *B. rapa* S8-mSRK complexed with S8-SP11, and compared the structure of this complex with the published structure of another SRK-SP11 pair, S9-SRK and S9-SP11 (Ma et al., 2016). S8-mSRK is a mutated form of S8-SRK, containing 11 amino acids different from S8-SRK, and it was used in this work, because it retained the ability to bind S8-SP11, but unlike S8-SRK, did not aggregate when

expressed in insect cells. Based on these crystal structures, the authors used computational modeling to predict the structures of five additional SRK/SP11 pairs of different S-haplotypes, and classified all seven pairs into two subgroups based on the mode of SRK/SP11 interactions.

I wish to first point out that the writing of this manuscript needs major improvement, as there are numerous instances of poor choice of words, missing articles, inappropriate use of articles, typographical errors, unclear sentences, and grammatically incorrect sentences. These writing issues make this manuscript a very difficult read. However, I will focus my comments below on the scientific merit of the manuscript, as I trust that if this manuscript were to be considered further by Nature Communications, the authors would be required to seek the help of professional English editors, or plant biologists with good English writing skills, to significantly improve their writing.

General Comments:

This is a nice piece of work, but as a similar crystal structure was reported by Ma et al. in 2016, this work must provide substantial new information to justify publication in Nature Communications. If the authors indeed have established a "universal" model of "self/non-self-discrimination" between SRK and SP11 in Brassica SI, then I would consider this accomplishment substantial. However, based on the data presented, I am not sure this is the case. Perhaps the authors fail to clearly articulate this accomplishment in the manuscript, which seems to be a compilation of overly detailed structural information. The authors' finding of two different modes of SRK/SP11 interactions between the S8 subgroup and S9 subgroup is based on the crystal structures of only two SRK-SP11 complexes (one of which was published by Ma et al.), and the results of computational modeling of SRK/SP11 interactions for five other S-haplotypes using these two crystal structures. The seven S-haplotypes the authors have examined are phylogenetically separated into two closely related subgroups (Supplementary Figure 7). It may not be surprising that the SRK-SP11 pairs in the same subgroup utilize a similar mode of interactions for self-recognition. However, in order to establish a "universal" model of "self/non-self-discrimination" between SRK and SP11, I would think that the authors should examine SRK-SP11 pairs of S-haplotypes that are phylogenetically distant from the S8 and S9 subgroups to see whether the modes of self/non-self-discrimination determined in this study would be applicable. This study would be of greater significance if the rule established from the S8- and S9-subgroups could be applied to SRK-SP11 pairs of S-haplotypes that are phylogenetically distant. In this regard, Ma et al. (2016) also used molecular docking to predict the structures of two SRK-SP11 pairs of *Arabidopsis lyrata* (Sa and S25 of *A. lyrata* shown in Supplementary Figure 7 of Ma et al. 2016), and showed that the "recognition mechanism" is similar to that of the S9-SRK/S9-SP11 pair for which they had determined the crystal structure. However, Ma et al. (2016) didn't examine the validity of their SP11 structure modeling as rigorously as the authors of this manuscript did. It would thus be interesting for the authors to use the improved methodology of the structure prediction presented in this manuscript to re-examine the results of Ma et al. (2016) to test whether the modes of self-recognition established in this work can be applied to phylogenetically distantly related S-haplotypes.

I have another major comment. Professor Takayama's own group previously showed that the soluble extracellular domain of S8-SRK alone did not exhibit high-affinity binding to biotin-labeled S8-SP11, thus suggesting a role of the transmembrane domain of SRK in its interaction with SP11 (Shimosato et al., *Plant Cell* 19, 107–117 2007). This work, as well as the work reported by Ma et al. (2016), only used the soluble extracellular domain of an SRK for structural determination, and did not examine the contribution of the transmembrane domain. The authors should have discussed the finding from this previous paper and its implications for the current work. In view of the result of this earlier work, I question the authors' claim that they have uncovered the detailed mechanism of self/non-self-discrimination in Brassica SI.

Specific Comments:

Abstract

Line 36: Throughout the manuscript, the authors use the term "SRK" to refer to the extracellular region (ectodomain) of S8-SRK, the extracellular region of S9-SRK reported by Ma et al. (2016), and the computationally predicted structures of the extracellular regions of the other five SRKs. This term is misleading, as only the structures of the extracellular regions of these SRKs are determined/predicted/compared. I would suggest that the authors replace "SRK" with "eSRK" (ectodomain of SRK), the term properly used by Ma et al. (2016).

Lines 40-41: The authors should briefly explain what they mean by "intra- and inter-subgroup combinations", as this information is important for the readers to understand the significance of their subsequent statement that the modes of SRK-SP11 interactions are different between these two subgroups.

Introduction

Line 55: I understand that two different names, SP11 and SCR, are used in the literature to designate the male determinant. For example, Ma et al. (2016) used SCR, and the authors use SP11 in this manuscript and in their previous publications. The authors should point out this naming issue in the Introduction, and when they refer to the male determination studied in Ma et al. (2016), they should not unilaterally change the name used by Ma et al., e.g., from SCR9 to S9-SP11. They should use SCR9, and not S9-SP11, when citing the results of Ma et al. (2016), and provide a statement to the effect that SCR9 is the same as S9-SP11 due to different naming schemes.

Lines 67-69: The authors seem to downplay the significance of Ma et al. (2016) when citing this paper. Considering that this paper reported the first crystal structure of an eSRK-SCR (SP11) complex, the authors should have mentioned the major findings from this paper, and justified the need for determining the crystal structure of another pair of eSRK-SP11. For example, is there any essential/key information that was missing/unclear from this earlier work? Or, perhaps the authors felt that having another structure would allow them to perform comparative studies to assess whether there are both common structural features and unique structural features between two different pairs of eSRK-SP11. Regardless, the authors should provide justification for their work in the Introduction.

Results

Lines 79-82: In the pull-down assay of S8-mSRK with biotin-labeled S8-SP11 shown in Figure 1b, only S8-mSRK fused with an HLH-ZIP dimerization domain can pull down S8-SP11. In the paper by Shimosato et al. (2007), Professor Takayama's group already showed that fusion of the HLH-ZIP domain is essential for eSRK to form a dimer in the in vitro pull-down assay, as the monomeric eSRK showed no binding with its self SP11 in vitro. The authors should have cited their earlier work, so that the readers can understand the reason for their fusing the HLH-ZIP domain to S8-mSRK in this work. Also, the authors should have discussed why S8-mSRK cannot bind S8-SP11 in the in vitro pull-down assay, but can form a complex with S8-SP11 in their crystal structure determination experiment, as well as in ITC, NMR and gel filtration experiments.

Line 89: The left-half of the parentheses in "Supplementary Fig. 2c)" is missing.

Line 91: "Brassica" is misspelled "Brassca".

Line 111: I believe that "Lys63 of S8-mSRK" should be "Lys63 of S8-SP11".

Line 112: What is the corresponding residue of Met64 of S8-SP11 in S9-SP11? Does the difference contribute to the observation that S8-SP11, but not S9-SP11, forms a dimer? This difference in the

ability to homodimerize is very interesting, and deserves more examination in this manuscript. For example, would mutating Met64 of S8-SP11 abolish its function in SI in the pollination bioassay?

Line 117: "steric crash" should be "steric clash".

Lines 134-135: In the pull-down assay shown in Figure 2g, when the authors set out to mutate the eight residues of S8-mSRK, how did they decide to which amino acid(s) each was to be mutated? Did they make the decision based on the corresponding amino acids of some other SRKs that do not interact with S8-SP11, or based on the biochemical properties of these amino acid residues? The authors should have provided a biological or biochemical rationale.

In Figure 2g, a weak band of biotin-tagged S8-SP11 is present in the "I339D (+)" lane, suggesting that, unlike other point mutations shown in this figure, mutation of I339D does not fully abolish the ability of S8-mSRK to interact with S8-SP11. Interestingly, Ma et al. (2016) also examined the amino acid residues surrounding T332 of S9-eSRK, which corresponds to I339 of S8-mSRK, and they found that simultaneously mutating T332 and the surrounding residues, H331 and R333, completely abolished the binding of S9-eSRK with S9-SP11. Therefore, it would be interesting for the authors to examine whether mutating the two residues surrounding I339 of S8-mSRK would completely abolish its interaction with S8-SP11.

Lines 142-143: In the "conservation analysis" shown in Supplementary Figure 3m, the authors should have mentioned the S-haplotypes used in this analysis. Also, the method for generating the conservation profile is not described in the manuscript.

Lines 159-160: Based on the crystal structure, the authors point out several residues of S8-mSRK that might be involved in its dimerization. It would be more convincing to experimentally test whether these residues of S8-mSRK are indeed critical for S8-mSRK dimerization, especially those that are not conserved in S9-SRK. For example, the authors could also use gel infiltration to test if mutating some of these residues of S8-mSRK to the corresponding residues in S9-eSRK would disrupt S8-mSRK dimerization, similar to the assay performed by Ma et al. (2016) and shown in Figure 5 of that paper. Moreover, Ma et al. (2016) confirmed the role of V211 and P294 in S9-eSRK dimerization. However, these residues are not indicated by blue circles in Supplementary Figure 5h.

Line 194: I think it is more appropriate to say "large negative deltaG values" than "small deltaG values". In Line 210, the authors use a similar term as I suggest here, but it is not clear whether they use the word "largely" in "largely negative deltaG values" to mean "mostly", or they use this word to intend to mean "large".

Lines 202-203: The authors should have pointed out that the term "topology" used here refers to the topology of the phylogenetic trees.

As S8-SP11, but not S9-SP11, is able to homodimerize, I was wondering whether the authors found that, in their computational modeling, all the other SP11s in the S8 subgroup were able to dimerize and all the other SP11s in the S9 subgroup were unable to do so.

Lines 233-234: I would suggest that the authors describe the exact percentage of sequence similarity/identity.

Line 237: It is not correct, and it is misleading, to say that Ser273 and Asp275 of S46-SRK are "mutated from" Asn271 and Glu272, respectively, of S8-SRK, as the differences in these amino acid residues between S46-SRK and S8-SRK simply reflect allelic variation of SRK.

Line 242: Is it possible that mutating these residues from the S8 version to the S46 version would enable the mutated SRK (i.e., S8-SP11-N231S E237D N337I) to interact with S46-SP11?

In the last lane of Figure 4b, a mutated S46-SRK with seven point mutations (four additional point mutations in addition to the three residues at the interface) was used for the pull-down assay, but the authors do not explain, in the text or figure legend, why this version of mutated S46-SRK was included in this assay.

Lines 245-250: Would a “higher order” mutant of S36-SP11 (i.e., S36-SP11-S36R K57R I58H) cause acceptance of S12 pollen grains by the S36S36 pistil treated with this mutant form of S36-SP11, just as treatment of the S36S36 pistil with S36-SP11-H62R caused acceptance of S12 pollen grains?

Line 248: The authors should have explained/described the basis of the bioassay, otherwise the readers would likely think that this is a self-pollination assay and would be confused by what the authors state here “... induce SI reaction against S36 pistil”. In this bioassay, a functional version of S36-SP11 will cause rejection of normally compatible pollen (S12 pollen here) when applied on an S36S36 pistil, whereas a non-functional S36-SP11 (i.e., S36-SP11-H62R), when applied on an S36S36 pistil, will result in the acceptance of S12 pollen by the S36S36 pistil. To help the readers better understand this bioassay, the authors should have clarified what crosses were performed, and clearly interpreted the pollination results.

A general comment for this section (“Self/nonself-discrimination in S8- and S9-subgroups”): Why for the S8-subgroup, was the site mutagenesis analysis performed on the SRK side, whereas for the S9-subgroup, the site mutagenesis analysis was performed on the SP11 side? The authors should explain.

Discussion

Line 263: Similar to my comments on Lines 202-203, the authors should have pointed out that the term “topology” refers to the topology of the phylogenetic trees.

Line 265: I would suggest that the authors perform selection analysis of SRK and/or SP11 (e.g., dN/dS analysis) to see whether there is positive selection on the residues that are presumed to have important function in self-recognition, in order to bolster the claim that some residues are “evolutionarily restricted”.

Methods

Line 285: “S8-SRK” should not be italicized as it refers to S8-SRK protein and not the gene encoding this protein. In the Methods section, the names of some other proteins are also incorrectly italicized (e.g., “S8-mSRK-HLH-ZIP” in Line 290).

Lines 288-290: This sentence is one of the many that are poorly written or grammatically incorrect. The authors may want to revise it to “S8-mSRK with eleven amino acid mutations ... was synthesized by step-by-step site-directed mutagenesis”. The authors should also describe the method they used to perform site-directed mutagenesis.

Line 333: The first sentence of this paragraph should be indented.

Line 376: I would suggest that “..infected Sf9 cells and cultured..” be changed to “..was used to infect Sf9 cells and the infected cells were cultured..”.

Line 400: The methods for several key computational analyses used in this work are not described, including the method for generating the phylogenetic tree shown in Supplementary Figure 7 and the method for generating the conservation profile of Brassica SRK shown in Supplementary Figure 3m.

Figures

Figure 3b: For the alignments shown in the figure, I would suggest that the authors indicate the S-haplotypes in the S8-subgroup and the S-haplotypes in the S9-subgroup, and perhaps separate these two subgroups with increased line spacing.

Figure 4d: In this figure, the authors should have pointed out the S-haplotype of the pistil (S36S36) and the S-haplotype of pollen (S12) used in the pollination bioassay.

Supplementary Figure 1a: The diagram of SRK is too small to see the exact positions of all the mutated amino acids, so some of the arrowheads are crowded together. In addition, the authors use the term "S8-SRK constructs" in the legend to this figure, but in fact only the eSRK is shown. The authors also use the same incorrect term in the legend to Supplementary Figure 1b. They should be more careful with the nomenclature, and should not confuse or mislead the readers.

Supplementary Figure 4a: In the alignment, what do the small yellow triangles indicate? These triangles are also used in the sequence alignment in Supplementary Figure 5h.

Supplementary Figure 7: The bootstrap values are hard to read.

Our responses to the reviewer's comments

General

We are grateful for the kind reviews of our manuscript. The changes we made in the text according to the reviewers' advice are marked by yellow highlighting. Following a reviewer's suggestion, here we used the term "eSRK", referring to the SRK ectodomain, to distinguish between the full-length protein and ectodomain of SRK. We added short summary of our work in the last of introduction and corrected the tense, terms and abbreviations in the text as following the journal guideline. The revised manuscript was edited by a scientific editor of English editing service. These changes are highlighted by cyan. We added six references and other six references were moved to Supplementary References.

Reviewer #1:

>The manuscript is well written, the introduction is informative, the experimental and theoretical approaches are sound and the methods section is complete. The authors present an interesting crystallographic work dealing with a very difficult problem. However, the figures could be improved to complement the text and the manuscript may benefit from a revision at several levels.

The differences on the overall structures (lines 91 to 105) and dimerization interface (lines 145 to 163) of S8-SRK-S8-SP11 and S9-SRK-S9-SP11 provide the structural basis for the self-discrimination ability. Consequently, they deserve a panel in figure 2. Conversely, the authors provide a very detailed description on the binding sites 1 and 2 that it is difficult to follow; this description needs to be more concise. In this direction, Figure 2 should be changed to compare side by side the details of S8-SRK-S8-SP11 and S9-SRK-S9-SP11 binding sites in the same orientation. The superimpositions shown in panels 2b,d and f confuse the reader.

According to the reviewer's suggestion, we moved Supplementary Fig. 3e, h, and i to Fig. 2 to allow easy comparison between the S₈-mSRK-S₈-SP11 and S₉-eSRK-S₉-SP11 structures. Along with those changes, we modified the labels of the figure panels and the legends of Fig. 2 and Supplementary Fig. 3. We also corrected the citations of Fig. 2 and Supplementary Fig. 3 in the text.

>The theoretical calculations validate well the conclusions drawn from the joined analysis of the crystal structures of S8-SRK-S8-SP11 and S9-SRK-S9-SP11. They provide elegant atomistic models and energy calculations that only account for the already predicted self/nonself-combinations. Indeed, this was pointed out in the paper by Ma et al. 2016. This should be made clear in the discussion section.

Reviewer #3 made a similar comment. To address these issues, we added the results of MM-GBSA experiments using models from another subgroup the class-II subgroup (S₂₉, S₄₀, S₄₄, and S₆₀; see Supplementary Fig. 7). We added a section to the end of Results, and also added data to Fig. 3, Table 1, Supplementary Fig 6e, and Supplementary Fig. 10. We

describe these changes in more detail in our response to Reviewer #3.

>The authors include 11 point mutations in the extracellular domain SRK to obtain functional protein for structural studies. The methods section would benefit from a description of the procedure that inspired these mutations.

Our preliminary domain expression experiments showed that the two lectin domains of S_8 -eSRK cause protein aggregation. To decrease the hydrophobic properties of S_8 -eSRK while maintaining the protein folding, we replaced their amino acids in the lectin domains with less hydrophobic amino acids from the same positions of S_8 -SLG or other SRK proteins. We added the following sentences to the Methods section.

“To suppress recombinant protein aggregation, the 11 amino acids were replaced with less hydrophobic residues of S_8 -SLG or other SRK proteins. The P79S, Y80E, and I81R mutations were derived from the S_8 -SLG sequence; F108V and L110R are from S_{12} -SRK; L180R is from S_{60} -SRK; F190S and L239S are from S_9 -SRK; and L214Q, V286G, and V287A are from S_{46} -SRK.” (page 15, line 377)

>The I/sI value in the highest resolution shell (5.38., supp. Table 1) suggests the diffraction data goes well beyond the 2.6 Å cutoff applied by the authors. Please provide an explanation in the methods section. Besides, the values of the CC1/2 and the R/R_{free} in the highest resolution shell should be also provided.

Even when we cut the diffraction data at 2.6-Å resolution, the R_{merge} value in the highest resolution was slightly high (0.63). Because 2.5-Å cutoff data were used, the R_{merge} value was over 0.8. Therefore, we decided to use the 2.6-Å cutoff. As suggested, we added CC1/2 and R_{work}/R_{free} values from the highest resolution to Supplementary Table 1.

>Line 188 “MM-GBSA (Molecular Mechanics-Generalized Born Surface Area)” should be written “Molecular Mechanics-Generalized Born Surface Area (MM-GBSA)”

Corrected as suggested. (page 9, line 206)

Reviewer #2:

>Did any of the 11 mutations to make the SRK8 protein stable correspond to the hypervariable regions I and II of SRK? Are these mutated sites conserved across haplotypes and whether this can influence inter and intra subgroup combination. Were any of these mutations located in the 31 contact position of SRK8 with S8-SP11? Authors need to explain the influence of these mutations on binding other SP11 haplotypes. Highlight the mutated residues in Supplementary figure 4.

We marked the positions of 11 mutations as blue boxes in Supplementary Fig. 4a, and added the following sentence to the legend:

“The positions of the eleven amino acid mutations in S_8 -mSRK are indicated by blue boxes.” (page 32, line 902)

Five of residues involved in the 11 S_8 -mSRK mutations (Y80E, F190S, L214Q, V286G, and V287A), are in contact to S_8 -SP11. L214Q is in HV I, and V286G and V287A are in HV II. Fortunately, these mutations do not have a major effect on S_8 -SP11 recognition. Some residues can be seen in the figures. Gln214 of S_8 -mSRK forms a hydrogen bond with Ser62 of S_8 -SP11 via a water molecule (Fig. 2a). Glu80 has weak van der Waals contacts with Thr41 and Thr42 of S_8 -SP11 (Fig. 2g). One of the two Ser190 residues in the S_8 -SP11– S_8 -mSRK complex has a hydrogen bond with Lys39 of S_8 -SP11 (Fig. 2g; Supplementary Fig. 3i). Low-energy contributions of Leu214, Val286, and Val287 residues in S_8 -SP11– S_8 -eSRK interaction can also be confirmed in Fig. 3c. We added the following sentence to the Results section:

“Although five of the eleven residues mutated in S_8 -mSRK contact S_8 -SP11, there is no negative effect on S_8 -SP11 recognition.” (page 7, line 147)

Reviewer #2 suggested that we investigate the binding effect of the 11 mutations in S_8 -mSRK against other haplotypes of SP11. We believe the comment might originate from a misunderstanding of the results in Fig. 3. To avoid artificial effects in the analysis of eSRK–SP11 interactions shown in Fig. 3, we reconstructed an S_8 -eSRK model from the crystal structure of S_8 -mSRK and used it for docking experiments with MM-GBSA. An insufficient explanation on our part might have led to a misunderstanding of the results. To prevent misunderstanding by the readers, we added the following sentence to the text.

“To avoid the artificial effects of the S_8 -mSRK mutations in the following experiments, we also made an S_8 -eSRK model from S_8 -mSRK.” (page 8, line 185)

>The authors claim that multiple residues in SRK8 are required for the tight binding of SRK8 with S_8 -SP11 (Lines 239 to 243, Fig. 4b). Contrarily in Fig. 2g, multiple single mutants are shown to abolish SRK8 binding to S_8 -SP11 including N271D mutation. Please address this discrepancy in the text.

In Fig. 2, we intentionally chose mutations that were significantly different in size and/or charge as those that lacked S_8 -SP11 recognition activity. Among them, N271D and I339D mutants still possessed weak binding ability. The results are thought to be due to the similar size of their side chains, even though their charges are different. On the other hand, the mutations used in Fig. 4 simply replaced the residues of S_8 -SRK with the residues from the same positions of S_{46} -SRK. These mutations are milder than those used in Fig. 2. N271S appears to be a weaker mutation than N271D, likely due to the smaller size and weaker charge of serine vs aspartic acid, resulting in weaker charge repulsion for the Ser271 against Asn65 in S_8 -SP11 (Fig. 2g). Thus, we added the following sentence in the text.

“In contrast to the result shown in Fig. 2j, this observation is the consequence of relatively mild differences in amino acid characteristics between S_8 -SRK and S_{46} -SRK.” (page 10, line 266)

Reviewer #3:

>I wish to first point out that the writing of this manuscript needs major improvement, as there are numerous instances of poor choice of words, missing articles, inappropriate use of articles, typographical errors, unclear sentences, and grammatically incorrect sentences. These writing issues make this manuscript a very difficult read. However, I will focus my comments below on the scientific merit of the manuscript, as I trust that if this manuscript were to be considered further by Nature Communications, the authors would be required to seek the help of professional English editors, or plant biologists with good English writing skills, to significantly improve their writing.

The manuscript has been reviewed by a scientific editor whose native language is English.

>This is a nice piece of work, but as a similar crystal structure was reported by Ma et al. in 2016, this work must provide substantial new information to justify publication in Nature Communications. If the authors indeed have established a “universal” model of “self/non-self-discrimination” between SRK and SP11 in *Brassica SI*, then I would consider this accomplishment substantial. However, based on the data presented, I am not sure this is the case. Perhaps the authors fail to clearly articulate this accomplishment in the manuscript, which seems to be a compilation of overly detailed structural information. The authors’ finding of two different modes of SRK/SP11 interactions between the S8 subgroup and S9 subgroup is based on the crystal structures of only two SRK-SP11 complexes (one of which was published by Ma et al.), and the results of computational modeling of SRK/SP11 interactions for five other S-haplotypes using these two crystal structures. The seven S-haplotypes the authors have examined are phylogenetically separated into two closely related subgroups (Supplementary Figure 7). It may not be surprising that the SRK-SP11 pairs in the same subgroup utilize a similar mode of interactions for self-recognition. However, in order to establish a “universal” model of “self/non-self-discrimination” between SRK and SP11, I would think that the authors should examine SRK-SP11 pairs of S-haplotypes that are phylogenetically distant from the S8 and S9 subgroups to see whether the modes of self/non-self-discrimination determined in this study would be applicable. This study would be of greater significance if the rule established from the S8- and S9-subgroups could be applied to SRK-SP11 pairs of S-haplotypes that are phylogenetically distant. In this regard, Ma et al. (2016) also used molecular docking to predict the structures of two SRK-SP11 pairs of *Arabidopsis lyrata* (S_a and S₂₅ of *A. lyrata* shown in Supplementary Figure 7 of Ma et al. 2016), and showed that the “recognition mechanism” is similar to that of the S9-SRK/S9-SP11 pair for which they had determined the crystal structure. However, Ma et al. (2016) didn’t examine the validity of their SP11 structure modeling as rigorously as the authors of this manuscript did. It would thus be interesting for the authors to use the improved methodology of the structure prediction presented in this manuscript to re-examine the results of Ma et al. (2016) to test whether the modes of self-recognition established in this work can be applied to phylogenetically distantly related S-haplotypes.

As we understand it, the crux of the reviewer’s comment is whether our findings regarding the mechanism of self/nonself-discrimination in *Brassica SI*, which we made by analyzing the S₈- and S₉-subgroups, can be adapted to other subgroups. To address this issue, although Reviewer #3 suggested that we use the *Arabidopsis lyrata* S_a and S₂₅ haplotypes, we chose class-II haplotypes (S₂₉, S₄₀, S₄₄, and S₆₀) for our subsequent analysis because this

manuscript focuses on *Brassica* SI (Supplementary Fig. 7). Because the haplotypes in the class-II subgroup are phylogenetically distant from the haplotypes in S_8 - and S_9 -subgroups among the known *Brassica* haplotypes, they are suitable for testing our hypothesis. We modeled class-II eSRK and SP11 proteins in a similar manner and analyzed their self- and nonself-interactions with MM-GBSA calculations. We added one section to the text to explain our analysis of the class-II subgroup, and also described the procedures in the Methods section. (page 11, line 281; page 19, line 503; page 20, line 538). We also added the figures and data in Fig. 3, Table 1, Supplementary Fig. 6, and Supplementary Fig. 10. The modeled class-II SP11 proteins formed defensin-like folds, similar to other models and crystal structures (Supplementary Fig. 6e). In the MM-GBSA analysis of class-II self- and nonself-complexes, the ΔG values of self-pair complexes were the most stable, suggesting that our model structures could strongly interact with their cognate partners (Fig. 3a).

The recognition modes of class-II haplotypes are similar to those of other members of the class-II subgroup, but not those of the S_8 - and S_9 -subgroups. For example, SRK proteins in class-II haplotypes have a specific four-residue insertion (FLNQ) in the HV I region. The Phe residue (i.e., Phe218 in S_{29}) among the insertion residues interacts with Phe/Tyr residue of SP11 (i.e., Phe75 in S_{29}) by aromatic–aromatic interaction (Fig. 3b, c; Supplementary Fig. 10). The ring portion of His/Phe62 residue of SP11 (position 30 in Fig. 3b) located on a cleft between HV I and HV III of eSRK, and the carbonyl group of the main chain in the same residue of SP11 forms hydrogen bonds with a Lys residue (i.e., K333 in S_{29}) in SRK-HV III (Supplementary Fig. 10). These residues are conserved in the class-II subgroup, but not in the S_8 - and S_9 -subgroups. Our contribution analysis of CR regions in class-II SP11 proteins also showed that the proportional contributions of CR regions to ΔG differ from those in the S_8 - and S_9 -subgroups (Table 1). Because the ranges of CR regions in class-II subgroup are slightly different from those of S_8 - and S_9 -subgroups, we re-defined the lengths of CR regions and the values of CR II in S_8 - and S_9 -subgroups were corrected (Table 1). Based on these observations, we believe that the recognition mode of the class-II subgroup does not conflict with our conclusion.

We also investigated how class-II haplotypes discriminate each other. For example, Ala58 in S_{44} -SP11 contacts Asp292 of S_{44} -eSRK. The corresponding residues of Ala58 in other haplotypes are Asp or Glu, which interfere with the Asp 292 of S_{44} -eSRK when incompatible SRK and SP11 proteins are docked. Conversely, the corresponding residues of Asp292 in other SRKs are Ala or Gly, which are smaller side chains suitable for interaction with Asp or Glu residues of cognate SP11 proteins (Supplementary Fig. 10). Val55 of S_{40} -SP11 contacts the hydrophobic region of the Arg82 side chain in S_{40} -eSRK. Because S_{29} -SP11 docked with S_{40} -eSRK, the corresponding residue S_{29} -SP11, Lys55, causes both steric hindrance and electric repulsion, which might contribute to S_{29}/S_{40} discrimination (Supplementary Fig. 10a, b). In the case of the S_{60} haplotype, Arg338 of S_{60} -eSRK forms a salt bridge with Glu58 of S_{60} -SP11, but the corresponding Gln residues in other class-II eSRK proteins cannot (Supplementary Fig. 10d). Tyr192 of S_{60} -eSRK (Ser in others) also forms methionine-aromatic interaction with Met80 of S_{60} -SP11, but not hydrophilic residues in other class-II SP11 proteins (Supplementary Fig. 10d). These S_{60} haplotype-specific interactions contribute to the self-specific stabilization in the MM-GBSA analysis.

Our results suggest that our method for analyzing self/nonself-discrimination is applicable to unrelated subgroups that have no suitable structural template, and the mechanism of self/nonself-discrimination identified in the analysis of S_8 - and S_9 -subgroups are common in other subgroups. We added the following sentence in the last paragraph of the Discussion section:

“We demonstrated that our methods can be used to analyze the interactions between SRK and SP11 in other unrelated subgroups and have potential applications for future analysis to identify unknown pairs of defensin-like ligands and SRK-like receptors.” (page 13, line 360)

>I have another major comment. Professor Takayama's own group previously showed that the soluble extracellular domain of S_8 -SRK alone did not exhibit high-affinity binding to biotin-labeled S_8 -SP11, thus suggesting a role of the transmembrane domain of SRK in its interaction with SP11 (Shimosato et al., *Plant Cell* 19, 107–117 2007). This work, as well as the work reported by Ma et al. (2016), only used the soluble extracellular domain of an SRK for structural determination, and did not examine the contribution of the transmembrane domain. The authors should have discussed the finding from this previous paper and its implications for the current work. In view of the result of this earlier work, I question the authors' claim that they have uncovered the detailed mechanism of self/non-self-discrimination in *Brassica* SI.

We would first point out that we have confirmed that the S_8 -SRK ectodomain and transmembrane domain (residues 30–468) can bind S_8 -SP11 (Shimosato et al., *Plant Cell*, 19, 107, 2007, Fig. 3). In our previous observation, S_8 -eSRK with transmembrane region (30–468) and S_8 -eSRK ectodomain (residues 30–443) fusions with HLH-ZIP bound S_8 -SP11, whereas S_8 -eSRK did not. We believe that this is due to a difference in SRK orientation. When SRK forms the heterotetramer complex, the first step is SP11 binding to the SP11 recognition site I of SRK, because the site I has a larger binding area than site II. Next, the SRK-SP11 heterodimer interacts with another SRK-SP11 heterodimer in a symmetric fashion to form the heterotetramer. In this process, the SRK-SP11 heterodimer has to meet a suitable partner in the proper orientation before the SP11 molecule is released from the binding SRK, as the SP11 and SP11 recognition site I of SRK still interact too weakly to form a stable complex. In the case of the SRK ectodomain containing transmembrane region, the SRK molecules are fixed on the membrane, and the orientations of all SRK molecules are almost perpendicular to the membrane plane. By contrast, when SRK ectodomains are in solution, they can be in any orientation, suggesting that it is more difficult for eSRK to meet a suitable partner than when it is on the membrane. In the case of the SRK-HLH-ZIP fusion, SRK molecules keep their partners nearby because HLH-ZIP protein forms a homodimer. Based on this viewpoint, we believe that eSRK has limited opportunity to meet its partner, reflected by the results of the pull-down experiment in Fig. 1b and our previous experiments. On the other hand, our ITC, NMR, and gel-filtration experiments, as well as the experiments by Ma et al., were performed under very high eSRK and SP11 concentrations. Under these conditions, the eSRK can find a partner much more easily than in the pull-down assay. In any case, these characteristics should be common in all SRK haplotypes, and we believe that this phenomenon is not directly related

to the mechanism of self/nonself-discrimination in *Brassica* SI. Therefore, we do not feel it is necessary to add the discussion above to the text.

>Line 36: Throughout the manuscript, the authors use the term “SRK” to refer to the extracellular region (ectodomain) of S8-SRK, the extracellular region of S9-SRK reported by Ma et al. (2016), and the computationally predicted structures of the extracellular regions of the other five SRKs. This term is misleading, as only the structures of the extracellular regions of these SRKs are determined/predicted/compared. I would suggest that the authors replace “SRK” with “eSRK” (ectodomain of SRK), the term properly used by Ma et al. (2016).

As suggested, we corrected “SRK” to “eSRK” in the text.

>Lines 40-41: The authors should briefly explain what they mean by “intra- and inter-subgroup combinations”, as this information is important for the readers to understand the significance of their subsequent statement that the modes of SRK-SP11 interactions are different between these two subgroups.

We added the definition, “(a group of phylogenetically neighboring haplotypes)”, after the words “intra- and inter-subgroup”. (page 2, line 41)

>Line 55: I understand that two different names, SP11 and SCR, are used in the literature to designate the male determinant. For example, Ma et al. (2016) used SCR, and the authors use SP11 in this manuscript and in their previous publications. The authors should point out this naming issue in the Introduction, and when they refer to the male determination studied in Ma et al. (2016), they should not unilaterally change the name used by Ma et al., e.g., from SCR9 to S9-SP11. They should use SCR9, and not S9-SP11, when citing the results of Ma et al. (2016), and provide a statement to the effect that SCR9 is the same as S9-SP11 due to different naming schemes.

To avoid confusing readers, we added the following text: “(SP11; also called SCR)” in page 3, line 57 and “(called eSRK9–SCR9 in this paper)” in page 3, line 71. Although Reviewer #3 asserted that we should not change the name used by Ma et al., we were the first to identify and name of S₉-SP11 (Suzuki et al., *Genetics*, 153, 391, 1999); this was the earliest paper to mention SP11/SCR. Moreover, the name of this protein in the NCBI database is also S₉-SP11 (accession No. BAA85458). We would point out that the unilateral changes of the name were made by Ma et al., *not* by us. The reviewer’s claim is not fair.

>Lines 67-69: The authors seem to downplay the significance of Ma et al. (2016) when citing this paper. Considering that this paper reported the first crystal structure of an eSRK-SCR (SP11) complex, the authors should have mentioned the major findings from this paper, and justified the need for determining the crystal structure of another pair of eSRK-SP11. For example, is there any essential/key information that was missing/unclear from this earlier work? Or, perhaps the authors felt that having another structure would allow them to perform comparative studies to assess whether there are both common structural features and unique

structural features between two different pairs of eSRK-SP11. Regardless, the authors should provide justification for their work in the Introduction.

The major achievements of Ma et al. is that they were the first to determine the eSRK–SP11 complex structure and revealed the mechanism by which S_9 -SRK recognizes the S_9 -SP11 molecule. However, they failed to clarify the mechanisms underlying self/nonself-discrimination and ligand-induced SRK dimerization. Indeed, Ma et al. stated in their own review article that “the structural mechanism underlying SCR-induced SRK homodimerization remains unknown” (Song et al., *Curr. Opin. Struct. Biol.*, 43, 18, 2017, page 24). Although Ma et al. (2016) also studied self/nonself-discrimination using homology-modeled structures and docking experiments, we believe Reviewer #3 understood after reading our manuscript that their results are unreliable because they used unverified models in their experiments. Out of a sense of collegiality, we elected not to describe their failure or unreliability in our manuscript. To make clear what is known and unknown, we corrected the last part of the Introduction as follows:

“The recent determination of the complex structure of the S_9 -SRK ectodomain (eSRK) and S_9 -SP11^{9,28} (called eSRK9–SCR9 in this paper) revealed the mechanism of S_9 -SP11 recognition by S_9 -SRK in *B. rapa* SI²⁹. However, the mechanisms of ligand recognition in other haplotypes and self/nonself-discrimination remained unknown.” (page 3, lines 70)

>Lines 79-82: In the pull-down assay of S8-mSRK with biotin-labeled S8-SP11 shown in Figure 1b, only S8-mSRK fused with an HLH-ZIP dimerization domain can pull down S8-SP11. In the paper by Shimosato et al. (2007), Professor Takayama’s group already showed that fusion of the HLH-ZIP domain is essential for eSRK to form a dimer in the in vitro pull-down assay, as the monomeric eSRK showed no binding with its self SP11 in vitro. The authors should have cited their earlier work, so that the readers can understand the reason for their fusing the HLH-ZIP domain to S8-mSRK in this work. Also, the authors should have discussed why S8-mSRK cannot bind S8-SP11 in the in vitro pull-down assay, but can form a complex with S8-SP11 in their crystal structure determination experiment, as well as in ITC, NMR and gel filtration experiments.

Our interpretation of these experiments is described above. According to the reviewer’s advice, we added the following sentences to the text.

“In the pull-down assay, S_8 -mSRK-HLH bound S_8 -SP11 but not S_8 -mSRK, consistent with our previous experiments¹⁸. S_8 -mSRK (S_8 -eSRK) seems difficult to form the ligand-receptor complex in the environment with low concentration of S_8 -mSRK such as the pull-down assay.” (page 5, lines 92)

>Line 89: The left-half of the parentheses in “Supplementary Fig. 2c)” is missing.

Line 91: “Brassica” is misspelled “Brassca”.

Line 111: I believe that “Lys63 of S8-mSRK” should be “Lys63 of S8-SP11”.

Line 117: “steric crash” should be “steric clash”.

We greatly appreciate the reviewer's assistance in identifying these typographical errors. We have made the appropriate corrections.

>Line 112: What is the corresponding residue of Met64 of S8-SP11 in S9-SP11? Does the difference contribute to the observation that S8-SP11, but not S9-SP11, forms a dimer? This difference in the ability to homodimerize is very interesting, and deserves more examination in this manuscript. For example, would mutating Met64 of S8-SP11 abolish its function in SI in the pollination bioassay?

Reviewer #3 suggested that S_8 -SP11 forms a homodimer before binding S_8 -mSRK. Our previous MS analysis of synthetic and native S_8 -SP11 showed a monomer peak, but no peak was observed around the mass of the dimer (Takayama et al., Nature, 413, 534, 2001, Fig. 1a). In addition, the retention time of the S_8 -SP11 peak in gel-filtration analysis indicates that the molecular weight is about 5 kDa (Supplementary Fig. 1c). These data suggest that almost all S_8 -SP11 molecules are monomers. Therefore, we do not feel the need to perform another experiment. Phe69 of S_9 -SP11, which corresponds to Met64 in S_8 -SP11, can be seen in Fig. 2c. Phe69 is surrounded by planes of Phe290, Phe267, and Phe189 of S_9 -SRK, and does not make contact with another molecule of S_9 -SP11. Ma et al. also pointed out that no S_9 -SP11 residue contacts another S_9 -SP11. In general, the contact area between two S_8 -SP11 molecules (52.1 \AA^2) is not sufficient to maintain the dimer without a covalent bond.

>Lines 134-135: In the pull-down assay shown in Figure 2g, when the authors set out to mutate the eight residues of S8-mSRK, how did they decide to which amino acid(s) each was to be mutated? Did they make the decision based on the corresponding amino acids of some other SRKs that do not interact with S8-SP11, or based on the biochemical properties of these amino acid residues? The authors should have provided a biological or biochemical rationale.

We first chose the residues of S_8 -mSRK that were important for S_8 -SP11 recognition based on detailed inspection of the S_8 -mSRK- S_8 -SP11 complex structure. Next, we changed the residues to have the opposite behavior, e.g., large vs. small, hydrophilic vs. hydrophobic, or positive vs. negative electrostatic charge. In many cases, because many residues were involved in protein-protein interactions, small changes in single residues were not sufficient to abolish the interaction. Based on this perspective, we chose mutations that had large differences in the features of their side chains. In the field of structural biology, this method is a common tool for confirming the trustworthiness of a determined structure and obtaining structural insight. Therefore, we do not provide a point-by-point explanation of each mutation in the text. In addition, we did not consider other SRK sequences or MD simulation data to select these mutations.

>In Figure 2g, a weak band of biotin-tagged S8-SP11 is present in the "I339D (+)" lane, suggesting that, unlike other point mutations shown in this figure, mutation of I339D does not fully abolish the ability of S8-mSRK to interact with S8-SP11. Interestingly, Ma et al. (2016) also

examined the amino acid residues surrounding T332 of S9-eSRK, which corresponds to I339 of S8-mSRK, and they found that simultaneously mutating T332 and the surrounding residues, H331 and R333, completely abolished the binding of S9-eSRK with S9-SP11. Therefore, it would be interesting for the authors to examine whether mutating the two residues surrounding I339 of S8-mSRK would completely abolish its interaction with S8-SP11.

As we mentioned in the response to Reviewer #2, the I339D result was the result of a relatively weak mutation; the side chains of isoleucine and aspartic acid are similar size. Reviewer #3 suggested that the corresponding residues of H331 and R333 in S₉-eSRK are potentially important in S₈-mSRK. However, the corresponding residues in S₈-mSRK, Glu338 and Ser340, do not contact S₈-SP11 (Supplementary Fig. 4a). Therefore, we believe that it is not necessary to perform mutational experiments on these residues.

>Lines 142-143: In the “conservation analysis” shown in Supplementary Figure 3m, the authors should have mentioned the S-haplotypes used in this analysis. Also, the method for generating the conservation profile is not described in the manuscript.

We used 30 SRK sequences derived from *B. rapa*. The method involved simply uploading the coordinate file and sequences to the web server. The result file can be opened in PyMOL. We do not feel this procedure needs to be described in the Methods section. We added information about the SRK sequences to the legend of Supplementary Fig. 3, as follows:

“Conservation scores, calculated with the ConSurf program using 30 *B. rapa* SRK sequences, are shown in color on the molecular surface of S₈-mSRK.” (page 32, line 890)

>Lines 159-160: Based on the crystal structure, the authors point out several residues of S8-mSRK that might be involved in its dimerization. It would be more convincing to experimentally test whether these residues of S8-mSRK are indeed critical for S8-mSRK dimerization, especially those that are not conserved in S9-SRK. For example, the authors could also use gel filtration to test if mutating some of these residues of S8-mSRK to the corresponding residues in S9-eSRK would disrupt S8-mSRK dimerization, similar to the assay performed by Ma et al. (2016) and shown in Figure 5 of that paper. Moreover, Ma et al. (2016) confirmed the role of V211 and P294 in S9-eSRK dimerization. However, these residues are not indicated by blue circles in Supplementary Figure 5h.

We believe that Reviewer #3 may have partially misunderstood the paper of Ma et al. Those authors reported S₉-eSRK residues involved in interactions with S₉-SP11 and SRK homodimerization (Ma et al., 2016, Fig. 4E). In the figure, V211 and P294 are marked as ligand recognition, but not SRK homodimerization. Because Ma et al. used the phrase “SCR-induced eSRK9 homodimerization” to explain the experiment involving V211 and P294 mutations, the words are referring to SRK–SP11 complex formation, not SRK homodimerization. Thus, Ma et al. tested the effect of mutations in eSRK9 residues involved in ligand recognition by observing complex formation. In the gel-filtration analysis, we can see SRK–SP11 complex formation, but not SRK dimerization, because SRK cannot form a homodimer without SP11. Therefore, it is difficult to assess residues

directly involved in SRK dimerization by gel-filtration analysis or other binding experiments.

>Line 194: I think it is more appropriate to say “large negative deltaG values” than “small deltaG values”. In Line 210, the authors use a similar term as I suggest here, but it is not clear whether they use the word “largely” in “largely negative deltaG values” to mean “mostly”, or they use this word to intend to mean “large”.

We corrected “very small ΔG values” to “large negative ΔG values” on line 194. On line 210, we meant “large”. We also corrected “largely negative ΔG values” to “large negative ΔG values” as suggested. (page 9, line 212; page 9, line 229)

>Lines 202-203: The authors should have pointed out that the term “topology” used here refers to the topology of the phylogenetic trees.

Corrected as suggested. (page 9, line 221)

>As S8-SP11, but not S9-SP11, is able to homodimerize, I was wondering whether the authors found that, in their computational modeling, all the other SP11s in the S8 subgroup were able to dimerize and all the other SP11s in the S9 subgroup were unable to do so.

As we mentioned above, we have no experimental evidence about dimer formation by S₈-SP11. In our modeled structures, no SP11 could form a dimer before interacting with SRK.

>Lines 233-234: I would suggest that the authors describe the exact percentage of sequence similarity/identity.

As suggested, we corrected as follows:

“Because the S₄₆-haplotype is closely related to S₈ in the S₈-subgroup (85% identity in eSRK, 36% in mature SP11), and the S₃₂- and S₃₆-haplotypes in the S₉-subgroup had high sequence identity (88% in eSRK, 75% in matured SP11), we examined the two pairs using computational and experimental analyses.” (page 10, line 253)

>Line 237: It is not correct, and it is misleading, to say that Ser273 and Asp275 of S46-SRK are “mutated from” Asn271 and Glu272, respectively, of S8-SRK, as the differences in these amino acid residues between S46-SRK and S8-SRK simply reflect allelic variation of SRK.

To avoid misleading the reader, we changed this sentence as follows:

“Our S₄₆-eSRK–S₄₆-SP11 complex model (Fig. 4a) revealed that Ser273 and Asp275 in S₄₆-eSRK, which are the residues corresponding to Asn271 and Glu273 in S₈-eSRK, respectively, were located close to Asn59 of S₄₆-SP11, and that Ile339, which has a more

hydrophilic side chain than the same position of S_8 -eSRK (Asn337), interacted with Phe39 in the α -helix of S_{46} -SP11.” (page 10, line 256)

>Line 242: Is it possible that mutating these residues from the S8 version to the S46 version would enable the mutated SRK (i.e., S8-SP11-N231S E237D N337I) to interact with S46-SP11?

We prepared streptavidin-tagged S_{46} -SP11 expressed in a bacterial expression system and tested the pull-down assay using S_8 -mSRK-HLH and S_8 -mSRK-HLH^{N271S,E273D,N337I} proteins. In our immunoblot analysis, both SRK protein bands were at the background level. Therefore, we believe that the mutations are not sufficient to give S_8 -eSRK the ability to recognize S_8 -eSRK.

>In the last lane of Figure 4b, a mutated S46-SRK with seven point mutations (four additional point mutations in addition to the three residues at the interface) was used for the pull-down assay, but the authors do not explain, in the text or figure legend, why this version of mutated S46-SRK was included in this assay.

We believe that Reviewer #3 is referring to the mutants of S_8 -mSRK. Because the N271S, E273D, and N337I mutations in S_8 -mSRK, which are sufficient to abolish the ability to recognize S_8 -SP11, are including in the seven-amino acid mutant, we thought that the readers would understand why the mutant also lost binding ability. To avoid confusion, we added the following sentence to the figure legend:

“ S_8 -mSRK-HLH^{N271S,E273D,N337I} and S_8 -mSRK-HLH^{N271S,E273D,N337I,E80G,S190P,Y198F,R367T} proteins lost the ability to bind S_8 -SP11.” (page 30, line 828)

>Lines 245-250: Would a “higher order” mutant of S36-SP11 (i.e., S36-SP11-S36R K57R I58H) cause acceptance of S12 pollen grains by the S36S36 pistil treated with this mutant form of S36-SP11, just as treatment of the S36S36 pistil with S36-SP11-H62R caused acceptance of S12 pollen grains?

We believe Reviewer #3 misunderstood the results shown in Fig. 4d. We administered S_{36} -SP11, S_{36} -SP11^{S36R}, S_{36} -SP11^{K57R,I58H}, and S_{36} -SP11^{H62R} to $S_{36}S_{36}$ pistils, and then pollinated with S_{12} pollen. If the treated proteins have S_{36} -SP11 activity, S_{12} pollen should be rejected on the $S_{36}S_{36}$ pistils. Only S_{36} -SP11^{H62R}-treated $S_{36}S_{36}$ pistil accepted S_{12} pollen tubes, which means that S_{36} -SP11^{H62R} cannot induce self-incompatibility response into $S_{36}S_{36}$ pistil. No S_{36} -SP11^{S36R,K57R,I58H} mutant was used in this study.

>Line 248: The authors should have explained/described the basis of the bioassay, otherwise the readers would likely think that this is a self-pollination assay and would be confused by what the authors state here “... induce SI reaction against S36 pistil”. In this bioassay, a functional version of S36-SP11 will cause rejection of normally compatible pollen (S12 pollen here) when applied on an S36S36 pistil, whereas a non-functional S36-SP11 (i.e., S36-SP11-H62R), when applied on an S36S36 pistil, will result in the acceptance of S12 pollen by the S36S36 pistil. To

help the readers better understand this bioassay, the authors should have clarified what crosses were performed, and clearly interpreted the pollination results.

To avoid confusing the reader, we changed the sentence as follows:

“To confirm the importance of the residues, we performed a pollination bioassay¹⁶. When an $S_{36}S_{36}$ pistil was treated with recombinant S_{36} -SP11 protein, compatible S_{12} pollen was rejected by S_{36} -SP11-induced SI reaction. Among the residues, only the S_{36} -SP11^{H62R} mutant did not induce the SI reaction against $S_{36}S_{36}$ pistils, suggesting that the mutation critically disrupted formation of the SRK-SP11 complex (Fig. 4d).” (page 11, line 2)

>A general comment for this section (“Self/nonself-discrimination in S8- and S9-subgroups”): Why for the S8-subgroup, was the site mutagenesis analysis performed on the SRK side, whereas for the S9-subgroup, the site mutagenesis analysis was performed on the SP11 side? The authors should explain.

Expression of functional eSRK is very difficult in any expression system. In the S_8 -subgroup, we established an expression system for S_8 -mSRK after 5 years of trial and error; that is why we chose that protein. By contrast, we could not obtain functional S_{32} - or S_{36} -eSRK in an insect cell system or by transient expression in *N. benthamiana*. Fortunately, we succeeded in expressing functional S_{36} -SP11 protein. Therefore, we used S_{36} -SP11 protein from the S_9 -subgroup.

>Line 263: Similar to my comments on Lines 202-203, the authors should have pointed out that the term “topology” refers to the topology of the phylogenetic trees.

Corrected as suggested. (page 13, line 347)

>Line 265: I would suggest that the authors perform selection analysis of SRK and/or SP11 (e.g., dN/dS analysis) to see whether there is positive selection on the residues that are presumed to have important function in self-recognition, in order to bolster the claim that some residues are “evolutionarily restricted”.

We calculated K_a/K_s values of SRK in the S_8 - (S_8 , S_{46} , S_{47} , and S_{61}) and S_9 -subgroups (S_9 , S_{32} , S_{36} , and S_{45}) separately because the positions of the conserved residues that recognize cognate SP11 differ between the subgroups. The important residue for SP11 recognition in S_8 -mSRK is Tyr275, which is located on the bottom of the SP11 binding pocket and interacts with Lys63 of S_8 -SP11 on the plane of aromatic ring and also interacts with Asn65 of another S_8 -SP11 molecule on the reverse side of the plane (Fig. 2a, g). The K_a/K_s value of the position of Tyr275 is 0.24, suggesting a strong restriction. Interestingly, the K_a/K_s value of the corresponding position in the S_9 -subgroup (Phe267 in S_9) is 2.0. Although Phe267 in S_9 -eSRK interacts with Phe69 of S_9 -SP11, the reverse side does not interact with another SP11 as in the S_8 -complex (Fig. 2c). Thus, the K_a/K_s value presumably reflects the differing importance of their ligand recognitions. In another case, Arg303 in S_8 -mSRK

forms three hydrogen bonds with S_8 -SP11 (Fig. 2d), and the K_a/K_s value of the position is 0.22. Because no other amino acids can form three hydrogen bonds like Arg, the position in S_8 -subgroup is likely restricted to Arg. By contrast, the corresponding residue Met295 in S_9 -eSRK contributes to S_9 -SP11 recognition through hydrophobic interaction (Fig. 2f); however, there is still some space between S_9 -eSRK and S_9 -SP11, and other hydrophobic side chains can be replaced in this position. The K_a/K_s value of the position in S_9 -subgroup is 1.6.

In the case of the S_9 -subgroup, one of the important positions for SP11 recognition is Phe189 in S_9 -eSRK, which contacts Pro68 and Phe69 of S_9 -SP11 through a hydrophobic interaction resulting closer position of the $\beta 2$ – $\beta 3$ loop of S_9 -SP11 against S_9 -eSRK than that of S_8 -complex (downward direction in Fig. 2b). Because the corresponding residue Tyr198 in S_8 -complex does not form a hydrogen bond with the neighboring Lys63 of S_8 -SP11, the contribution to the ligand–receptor interaction is not as important. Thus, the K_a/K_s values are 0.32 at Phe189 of S_9 -eSRK in the S_9 -subgroup, and 0.99 at the corresponding position in the S_8 -subgroup. Lys206 in S_9 -eSRK also contributes to the interaction with the $\beta 2$ – $\beta 3$ loop of S_9 -SP11 near Phe189 by forming a hydrogen bond (Fig. 2c). The corresponding residue in S_8 -mSRK is Gln214, which forms a hydrogen bond with SP11 via a water molecule. Gln214 is one of the 11 mutations introduced in S_8 -mSRK (Supplementary Fig. 1); the original amino acid is leucine. When Gln214 is replaced by Leu, the hydrogen bond will be broken, creating a space between ligand and receptor. In this case, the K_a/K_s values are 0.32 in the S_9 -subgroup and 1.1 in the S_8 -subgroup. These observations suggest that some key residues required for ligand recognition are evolutionarily restricted. However, all such residues are not restricted. For example, I339 in S_8 -mSRK and D330 in S_9 -eSRK are involved in ligand recognition, but the K_a/K_s values at these positions are 0.98 and 1.9 (Fig. 2d, f). In our observations, because the residues have a large space around them, the residues tend to be not restricted. In the case of I339 in S_8 -mSRK, other hydrophobic amino acids can also maintain a stable hydrophobic interaction with Phe34 of S_8 -SP11.

Although the K_a/K_s analysis provided interesting insight into the evolution of these subgroups, we did not discuss these data in the text because the sample numbers are too small. To make the discussion more accurate, we changed the sentence as follows:

“Some key residues important for self-recognition within each subgroup appear to be evolutionarily restricted, because mutations in these residues are more likely than changes in minor residues to abolish recognition ability, as in the case of other highly conserved amino acids required for protein function.” (page 13, line 348)

>Line 285: “S8-SRK” should not be italicized as it refers to S8-SRK protein and not the gene encoding this protein. In the Methods section, the names of some other proteins are also incorrectly italicized (e.g., “S8-mSRK-HLH-ZIP” in Line 290).

Corrected as suggested. (page 15, lines 370–387)

>Lines 288-290: This sentence is one of the many that are poorly written or grammatically incorrect. The authors may want to revise it to “S8-mSRK with eleven amino acid mutations ...

was synthesized by step-by-step site-directed mutagenesis". The authors should also describe the method they used to perform site-directed mutagenesis.

We corrected as follows:

"*S₈-mSRK*, which encodes *S₈-eSRK* containing 11 amino acid mutations (P79S, Y80E, I81R, F108V, L110R, L180R, F190S, L239S, L214Q, V286G, and V287A), was synthesized by step-by-step site-directed mutagenesis using the KOD -Plus- Mutagenesis Kit (TOYOBO)." (page 15, line 374)

>Line 333: The first sentence of this paragraph should be indented.

Corrected as suggested. (page 16, line 420)

>Line 376: I would suggest that "..infected Sf9 cells and cultured.." be changed to "..was used to infect Sf9 cells and the infected cells were cultured..".

Corrected as suggested. (page 18, line 466)

>Line 400: The methods for several key computational analyses used in this work are not described, including the method for generating the phylogenetic tree shown in Supplementary Figure 7 and the method for generating the conservation profile of Brassica SRK shown in Supplementary Figure 3m.

We added the method used for phylogenetic tree generation in the Method section and added two references. As described above, we did not add the method for the conservation profile. (page 21, line 556)

>Figure 3b: For the alignments shown in the figure, I would suggest that the authors indicate the S-haplotypes in the S8-subgroup and the S-haplotypes in the S9-subgroup, and perhaps separate these two subgroups with increased line spacing.

Corrected as suggested.

>Figure 4d: In this figure, the authors should have pointed out the S-haplotype of the pistil (S36S36) and the S-haplotype of pollen (S12) used in the pollination bioassay.

We added this information to Fig. 4d.

>Supplementary Figure 1a: The diagram of SRK is too small to see the exact positions of all the mutated amino acids, so some of the arrowheads are crowded together. In addition, the authors use the term "S8-SRK constructs" in the legend to this figure, but in fact only the eSRK is shown. The authors also use the same incorrect term in the legend to Supplementary Figure 1b. They

should be more careful with the nomenclature, and should not confuse or mislead the readers.

We added the information about the positions of the mutations in Supplementary Fig. 1a, and we corrected Supplementary Fig. 1b as suggested. The legends of Supplementary Fig. 1a and b were changed as follows:

“**a**, Left, schematic diagram of S_8 -eSRK constructs. Arrowheads show the positions of mutations used in S_8 -mSRK. Right, list of mutations used for screening of S_8 -eSRK expression shown in b. **b**, Screening of S_8 -eSRK constructs for stable overexpression in Sf9 cells. Numbers in the construct names indicate that the constructs have the corresponding mutations listed in a.” (page 30, line 841)

>Supplementary Figure 4a: In the alignment, what do the small yellow triangles indicate? These triangles are also used in the sequence alignment in Supplementary Figure 5h.

The yellow triangles show the positions of S_8 -SRK. We added position numbers to Supplementary Fig. 4a and Supplementary Fig. 5h. We also added the following sentence in both Figures:

“Yellow arrowheads and upper numbers show the positions of S_8 -SRK.” (page 32, line 897; page 33, line 916)

>Supplementary Figure 7: The bootstrap values are hard to read.

These have been corrected.

REVIEWERS' COMMENTS:

Reviewer #1 (Remarks to the Author):

The authors have address most of my concerns. However, still it is not clear why the authors select 2.6 Å as a resolution cut-off for crystallographic refinement. This is usually based on $CC1/2 > 0.5$ or $I/s > 1.5$. Authors should provide their criteria and explain it clearly in the methods section.

Reviewer #2 (Remarks to the Author):

The authors have convincingly addressed my concerns.

Reviewer #3 (Remarks to the Author):

In my review of the authors' previous submission, I felt that, although they did a nice job in accomplishing a difficult task of structural determination of eSRK/SP11 complexes, the way the manuscript was prepared didn't do their work justice. Thus, I provided a comprehensive and detailed review, with constructive comments, to help the authors improve the quality of the manuscript. I am pleased to see that the authors have made a conscientious effort to address all my comments, point-by-point, during the manuscript revision. For the experiments that I suggested that the authors do, they have done some of them, and provide sound reasons for why they chose not to do the others. Also, as I suggested, they sought out a scientific editor whose native language is English to extensively edit their manuscript. All in all, I believe that this manuscript is much improved from the previous submission.

I have the following largely minor comments on the revised manuscript.

Lines 57 and 71: The authors seem to take issue with my comment that they use the name "S9-SP11" when mentioning "S9-SCR", the name used in Ma et al. (2016), without providing any explanation. My comment stemmed from the concern that readers may be confused by the different names, S9-SP11 and S9-SCR, used by the authors and Ma et al. to indicate the same allelic variant of the male determinant, as I would think that most readers are not aware of the historical account of the identification and naming of the male determinant gene the authors provide in their response. Thus, my comment has nothing to do with which group identified and named this gene first, and in fact I had no idea about the history the authors describe. In the revision, the authors have added the statements that SP11 is also named SCR in Line 57, and that S9-SP11 is also called S9-SCR in Line 71. This clarification is precisely what I asked the authors to do! However, in Line 71, it is unclear to me what paper the authors refer to in saying "(called S9-SCR in this paper)". If it is the paper by Ma et al. (2016), they should say "(called S9-SCR in Ma et al. [2016])".

Line 74: As this is the first time that S8-haplotype is mentioned, I would suggest that the authors point out that this haplotype, like S-9-haplotype studied by Ma et al. (2016), is also in *Brassica rapa*, so that readers know that the authors studied the crystal structure of an eSRK-SP11 complex in the same species as S9-eSRK-S9-SCR9/SP11 studied by Ma et al. (2016). This is also the first time that S8-eSRK-S8-SP11 is introduced, so I would suggest that the authors make clear that they actually examined a mutant form of S8-eSRK with 11 amino acids different from wild-type. In Line 87, the authors name this mutant form of S8-eSRK as S8-mSRK, but this name might give readers the wrong impression that S8-mSRK contains a different domain of SRK than S9-eSRK. Perhaps, it would be less confusing to name the mutant form of S8-eSRK as S8-emSRK or S8-meSRK?

Line 92: I don't think the authors intend to say that "... S8-mSRK-HLH bound S8-SP11, but not S8-mSRK,". They should change this part of the sentence to "... S8-mSRK-HLH, but not S8-mSRK, bound S8-SP11,"

Lines 93-95: The authors explain why S8-mSRK failed to bind S8-SP11 in the pull-down assay (because the concentration of S8-mSRK was too low); however, they do not explain why HLH-ZIP-fused S8-mSRK could bind S8-SP11 in this assay. They provide a clear explanation in their response letter (due to the ability of HLH-ZIP to form dimers, thus increasing the ability of S8-mSRK to contact S8-SP11). I would suggest that the authors include this explanation here, as it will benefit readers not familiar with the biochemical properties of HLH-ZIP. In addition, the authors state in the response letter that they could detect binding of S8-mSRK to S8-SP11 in ITC, gel-infiltration and NMR experiments because these experiments were performed at high concentrations of S8-mSRK and S8-SP11. This information should also be included here.

Lines 98-101: The sentence beginning in Line 98 is not clear to me.

Line 181: I think the term "Comprehensive analysis" is more appropriate than "Global analysis" in describing the various analyses reported in this section. Moreover, I would suggest that the authors change "eSRK-SP11 interaction" to "eSRK-SP11 interactions in class-I haplotypes" to emphasize that all eSRK-SP11 complexes they chose to model in this section belong to class-I haplotypes and to differentiate the results from those described in a latter section titled "Computational analysis in class-II haplotypes". In addition, I would suggest that, in the figures and tables where S8- and S9-subgroups are mentioned, the authors point out that both subgroups are in class-I.

Line 217: For the title of this section, I would suggest that the authors add "in the S8- and S9-subgroups", just as they do in the title of the next section (Line 251), to make clear that class-II eSRK-SP11 interactions are not described in this section. These interactions are described in a latter section clearly titled "Computational analysis in class-II haplotypes".

Lines 533-534: Some of the symbols and characters are not legible.

Our responses to the reviewer's comments

Reviewer #1:

>The authors have address most of my concerns. However, still it is not clear why the authors select 2.6 Å as a resolution cut-off for crystallographic refinement. This is usually based on $CC1/2 > 0.5$ or $I/\sigma > 1.5$. Authors should provide their criteria and explain it clearly in the methods section.

As suggested, we added the cut off criteria in the Methods section as following:
“We cut off the native data at 2.6 Å resolution due to the high R_{merge} value (0.8 >), even though the $CC1/2$ and $I/\sigma I$ values were still enough.” (page 15, line 421)

Reviewer #2:

>The authors have convincingly addressed my concerns.

Reviewer #3:

>Lines 57 and 71: The authors seem to take issue with my comment that they use the name “S9-SP11” when mentioning “S9-SCR”, the name used in Ma et al. (2016), without providing any explanation. My comment stemmed from the concern that readers may be confused by the different names, S9-SP11 and S9-SCR, used by the authors and Ma et al. to indicate the same allelic variant of the male determinant, as I would think that most readers are not aware of the historical account of the identification and naming of the male determinant gene the authors provide in their response. Thus, my comment has nothing to do with which group identified and named this gene first, and in fact I had no idea about the history the authors describe. In the revision, the authors have added the statements that SP11 is also named SCR in Line 57, and that S9-SP11 is also called S9-SCR in Line 71. This clarification is precisely what I asked the authors to do! However, in Line 71, it is unclear to me what paper the authors refer to in saying “(called S9-SCR in this paper)”. If it is the paper by Ma et al. (2016), they should say “(called S9-SCR in Ma et al. [2016])”.

We corrected “(called eSRK9–SCR9 in this paper)” to “(called eSRK9–SCR9 in Ma et al. [2016])” as suggested. (Page 3, line 74)

>Line 74: As this is the first time that S8-haplotype is mentioned, I would suggest that the authors point out that this haplotype, like S-9-haplotype studied by Ma et al. (2016), is also in *Brassica rapa*, so that readers know that the authors studied the crystal structure of an eSRK-SP11 complex in the same species as S9-eSRK-S9-SCR9/SP11 studied by Ma et al. (2016). This is also the first time that S8-eSRK-S8-SP11 is introduced, so I would suggest that the authors make clear that they actually examined a mutant form of S8-eSRK with 11 amino acids different from wild-type. In Line 87, the authors name this mutant form of S8-eSRK as S8-mSRK, but this name might give readers the wrong impression that S8-mSRK contains a different domain of SRK than S9-eSRK. Perhaps, it would be less confusing to name the mutant form of S8-eSRK as S8-emSRK or S8-meSRK?

Following the reviewer's advice, we corrected "Here, we report the crystal structure of S₈-eSRK-S₈-SP11 complex, ..." to "Here, we report the crystal structure of engineered S₈-eSRK and S₈-SP11 complex derived from S₈-haplotype in *B. rapa*, ...". (page 3, line 76)

We also corrected the term "S₈-mSRK" to "S₈-meSRK" in the text and figures.

>Line 92: I don't think the authors intend to say that "... S8-mSRK-HLH bound S8-SP11, but not S8-mSRK,". They should change this part of the sentence to "... S8-mSRK-HLH, but not S8-mSRK, bound S8-SP11,"

As suggested, we corrected "S₈-meSRK-HLH bound S₈-SP11 but not S₈-meSRK," to "S₈-meSRK-HLH, but not S₈-meSRK, bound S₈-SP11,.". (page 5, line 95)

>Lines 93-95: The authors explain why S8-mSRK failed to bind S8-SP11 in the pull-down assay (because the concentration of S8-mSRK was too low); however, they do not explain why HLH-ZIP-fused S8-mSRK could bind S8-SP11 in this assay. They provide a clear explanation in their response letter (due to the ability of HLH-ZIP to form dimers, thus increasing the ability of S8-mSRK to contact S8-SP11). I would suggest that the authors include this explanation here, as it will benefit readers not familiar with the biochemical properties of HLH-ZIP. In addition, the authors state in the response letter that they could detect binding of S8-mSRK to S8-SP11 in ITC, gel-infiltration and NMR experiments because these experiments were performed at high concentrations of S8-mSRK and

S8-SP11. This information should also be included here.

As suggested, we changed the explanation “..., S₈-meSRK (S₈-eSRK) seems difficult to form the ligand-receptor complex in the environment with low concentration of S₈-meSRK such as the pull-down assay.” to “..., S₈-meSRK (S₈-eSRK) seems difficult to form the ligand-receptor complex in the environment with low concentration of S₈-meSRK such as the pull-down assay, in contrast to the high concentration conditions in ITC, gel-filtration, and CSP experiments. Dimerization domain (HLH) in S₈-meSRK-HLH is supposed to enhance S₈-meSRK–S₈-SP11 interaction by supporting the SP11–induced SRK dimerization.” (page 5, line 97)

>Lines 98-101: The sentence beginning in Line 98 is not clear to me.

It may be unclear. We changed the part “The two molecules in the complex are almost the same in both S₈-meSRK and S₈-SP11, ...” to “The structures of symmetrical molecules in a single complex are almost the same in both S₈-meSRK and S₈-SP11, ...” (page 5, line 105)

>Line 181: I think the term “Comprehensive analysis” is more appropriate than “Global analysis” in describing the various analyses reported in this section. Moreover, I would suggest that the authors change “eSRK-SP11 interaction” to “eSRK-SP11 interactions in class-I haplotypes” to emphasize that all eSRK-SP11 complexes they chose to model in this section belong to class-I haplotypes and to differentiate the results from those described in a latter section titled “Computational analysis in class-II haplotypes”. In addition, I would suggest that, in the figures and tables where S8- and S9-subgroups are mentioned, the authors point out that both subgroups are in class-I.

As suggested, we changed the subheading “Global analysis of eSRK–SP11 interaction” to “Comprehensive analysis of class-I eSRK–SP11 interactions”. (page 8, line 188)

We also changed the term “global” to “comprehensive” (page 4, line 79; page 9, line 218) and a sentence “Homology modeling of eSRK structures ...” to “Homology modeling of eSRK structures, which belong to class-I haplotypes, ...” (page 8, line 190).

We added the information “class-I” to Fig. 3 and Table 1.

>Line 217: For the title of this section, I would suggest that the authors add “in the S₈- and S₉-subgroups”, just as they do in the title of the next section (Line 251), to make clear that class-II eSRK-SP11 interactions are not described in this section. These interactions are described in a latter section clearly titled “Computational analysis in class-II haplotypes”.

As suggested, we changed the subheading “Different modes of SP11 recognition” to “Different modes of SP11 recognition in the S₈- and S₉-subgroups”. (page 9, line 223)

>Lines 533-534: Some of the symbols and characters are not legible.

We corrected. (page 19, lines 537–538)